# Utility of red-light ultrafast optogenetic stimulation of the auditory pathway

Burak Bali[1,2,3], David Lopez de la Morena[1,2,4,5,†] (iD), Artur Mittring[1,6,†], Thomas Mager[1,7],
Vladan Rankovic[1,3,*] (iD), Antoine Tarquin Huet[1,4,6,7,**] (iD) & Tobias Moser[1,4,5,7,***] (iD)

## Abstract

Optogenetic stimulation of spiral ganglion neurons (SGNs) in the ear provides a future alternative to electrical stimulation used in current cochlear implants. Here, we employed fast and very fast variants of the red-light-activated channelrhodopsin (ChR) Chrimson (f-Chrimson and vf-Chrimson) to study their utility for optogenetic stimulation of SGNs in mice. The light requirements were higher for vf-Chrimson than for f-Chrimson, even when optimizing membrane expression of vf-Chrimson by adding potassium channel trafficking sequences. Optogenetic time and intensity coding by single putative SGNs were compared with coding of acoustic clicks. vf-Chrimson enabled putative SGNs to fire at near-physiological rates with good temporal precision up to 250 Hz of stimulation. The dynamic range of SGN spike rate coding upon optogenetic stimulation was narrower than for acoustic clicks but larger than reported for electrical stimulation. The dynamic range of spike timing, on the other hand, was more comparable for optogenetic and acoustic stimulation. In conclusion, f-Chrimson and vf-Chrimson are promising candidates for optogenetic stimulation of SGNs in auditory research and future cochlear implants.

**Keywords** channelrhodopsin; cochlear implant; dynamic range; gating; spiral ganglion; temporal coding

**Subject Categories** Biotechnology & Synthetic Biology; Neuroscience

## Introduction

Optogenetics, the control of cells with light, has revolutionized the life sciences and bears potential for innovative therapies such as for sensory restoration (Sahel & Roska, 2013; Dombrowski *et al*, 2019; Dieter *et al*, 2020; Kleinlogel *et al*, 2020). AAV-mediated optogenetics for vision restoration has recently entered clinical trials (ClinicalTrials.gov identifier: NCT02556736, Allergan; NCT03326336, GenSight Biologics), while the development of optogenetic hearing restoration is at the preclinical stage. Restoring light sensitivity of the retina by optogenetics seems plausible and can likely be achieved with ChRs that deactivate within several milliseconds (Busskamp *et al*, 2012). The unmet need for means to restore vision is very high: Except for gene therapy of Leber's congenital amaurosis 2 by Luxturna, an FDA-approved AAV therapy, currently there are no treatment options for retinal degeneration. As a matter of fact, production of retinal implants has recently stopped in Europe and the United States.

Optogenetic hearing restoration, on the other hand, seems less intuitive, and rehabilitation of the deaf has successfully built on the well-established cochlear implant (CI). The CI enables open speech comprehension in most of the more than 700,000 users and is, therefore, considered the most successful neuroprosthesis (Zeng, 2017; Lenarz, 2018). However, an urgent need for further improvement of the CI remains as follows: Users typically have difficulty to understand speech in the presence of background noise and interpret the emotional tone in speech or appreciate music. As for the retinal implant, the biggest bottleneck of the CI is the widespread of current around each electrode contact, which limits the spectral resolution of sound coding (Kral *et al*, 1998). Using light for stimulation in future optical CIs (oCI) is one of the present developments to improve spectral coding by CIs, as light can be better spatially confined than electric current (Richter *et al*, 2011; Hernandez *et al*, 2014; Dieter *et al*, 2019, 2020; Keppeler *et al*, 2020). However, unlike for vision restoration, temporal fidelity of bionic sound coding is as important as spectral resolution of coding that is offered by optical stimulation. Therefore, fast-closing ChRs have been employed, such as Chronos (Duarte *et al*, 2018; Keppeler *et al*, 2018) and fast (f-) Chrimson (Mager *et al*, 2018; Huet *et al*, 2021), which enabled SGN firing at

1   Institute for Auditory Neuroscience and InnerEarLab, University Medical Center Göttingen, Göttingen, Germany
2   Göttingen Graduate School for Neurosciences and Molecular Biosciences, University of Göttingen, Göttingen, Germany
3   Restorative Cochlear Genomics Group, Auditory Neuroscience and Optogenetics Laboratory, German Primate Center, Göttingen, Germany
4   Auditory Neuroscience and Optogenetics Laboratory, German Primate Center, Göttingen, Germany
5   Auditory Neuroscience Group, Max-Planck-Institute for Experimental Medicine, Göttingen, Germany
6   Auditory Circuit Lab, Institute for Auditory Neuroscience and InnerEarLab, University Medical Center Göttingen, Göttingen, Germany
7   Cluster of Excellence "Multiscale Bioimaging: from Molecular Machines to Networks of Excitable Cells" (MBExC), University of Göttingen, Göttingen, Germany
    *Corresponding author. Tel: +49 176 21784294; E-mail: vladan.rankovic@gmail.com
    **Corresponding author. Tel: +49 551 39 22604; E-mail: antoine.huet@med.uni-goettingen.de
    ***Corresponding author. Tel: +49 551 39 63071; E-mail: tmoser@gwdg.de
    †These authors contributed equally to this work

near-physiological firing rates. The deactivation time constants of Chronos, very fast (vf)-Chrimson, and f-Chrimson at physiological temperature amounted to 0.8, 1.6, and 3.2 ms (Keppeler *et al*, 2018; Mager *et al*, 2018). Optogenetic SGN stimulation mediated by Chronos required highly potent vectors (Duarte *et al*, 2018; Keppeler *et al*, 2018) and optimization of trafficking (Keppeler *et al*, 2018). This indicates that short channel lifetimes need to be offset by high levels of plasma membrane expression for comparable photo-depolarization at the same light intensity.

f-Chrimson, on the other hand, mediated large photocurrents and enabled optogenetic stimulation of the auditory pathway at lower light intensities (Mager *et al*, 2018) than those required for Chronos (Keppeler *et al*, 2018). In fact, the radiant flux threshold for evoking auditory brainstem responses (oABRs) was an order of magnitude lower (0.5 mW) than for trafficking-optimized Chronos (6.6 mW vs. 14 mW for the original Chronos). Chrimson variants, as red-light-activated ChRs, have the additional advantage of neural stimulation with lower risk of phototoxicity and, therefore, are good candidates for translating optogenetic sensory restoration to the clinic (Kleinlogel *et al*, 2020). Here, we studied the utility of f-Chrimson and vf-Chrimson for optogenetic SGN stimulation addressing temporal fidelity of firing and intensity coding with reference to acoustic stimulation.

# Results

## Establishing, characterizing, and optimizing the utility of vf-Chrimson-mediated SGN stimulation

*In vitro* biophysical characterization (Mager *et al*, 2018) had indicated vf-Chrimson as a strong candidate for optogenetic stimulation of SGNs with high temporal fidelity. As the closing kinetics of vf-Chrimson is twice as fast as that of f-Chrimson (Mager *et al*, 2018), we anticipated an improved temporal fidelity of SGN stimulation. However, we reasoned that the trafficking of vf-Chrimson to the plasma membrane is less efficient, as the photocurrent density estimated in NG cells (neuroblastoma and neuroglioma cell line, NG108-15) for vf-Chrimson was about one fourth of that of f-Chrimson (Mager *et al*, 2018). Here, we compared expression of f-Chrimson and vf-Chrimson in NG108-15 cells using semiquantitative immunohistochemistry in three cultures each, with transfections, staining and imaging performed strictly in parallel. Using line profile analysis of eYFP immunofluorescence in confocal sections indicated three categories of cells (Fig EV1): i) cells with a clear plasma membrane peak of immunofluorescence (Fig EV1A and B), ii) cells with comparable immunofluorescence of plasma membrane and intracellular space (Fig EV1C and D), and iii) intracellular immunofluorescence outweighing that of the plasma membrane (Fig EV1E and F). The expression of f-Chrimson led to a larger fraction of NG108-15 cells with clear plasma membrane expression (9 out of 30 cells for f-Chrimson vs. three out of 30 cells for vf-Chrimson). Poorer plasma membrane expression likely explains why the maximal photocurrent density mediated by vf-Chrimson was lower than for f-Chrimson (Mager *et al*, 2018). Therefore, we aimed to improve membrane targeting by adding ER export and trafficking signals, isolated from a vertebrate inward rectifier potassium channel Kir2.1 (Stockklausner *et al*, 2001; Hofherr *et al*, 2005) sandwiching eYFP (Fig 1A). These sequences, nick-named ES (Export Signal) and TS (Trafficking

Signal), were previously shown to enhance the plasma membrane expression of opsins (Gradinaru *et al*, 2010; Keppeler *et al*, 2018).

Next, we analyzed the expression of vf-Chrimson-ES/TS and vf-Chrimson in mouse SGNs *in vivo*. We used intracochlear injection on postnatal day 6 (P6) of AAV-PHP.B (vf-Chrimson) or AAV-PHP.eB (vf-Chrimson-ES/TS) as highly potent viral vectors (Deverman *et al*, 2016; Chan *et al*, 2017; Keppeler *et al*, 2018) with similar titers, $8.7 \times 10^{12}$ and $1.1 \times 10^{13}$ genome copies/ml, respectively. We employed the human synapsin promoter (hSyn, Fig 1A) that drives efficient and specific channelrhodopsin expression in SGNs (Hernandez *et al*, 2014; Keppeler *et al*, 2018, 2020; Mager *et al*, 2018; Wrobel *et al*, 2018; Dieter *et al*, 2019; Huet *et al*, 2021). In order to evaluate viral transduction and membrane expression of vf-Chrimson-ES/TS and vf-Chrimson in SGNs, we performed confocal imaging of mid-cochlear sections following decalcification and immunolabeling for parvalbumin (PV, SGN marker) and GFP (for detection of the eYFP-tagged opsins).

First, we probed for plasma membrane targeting of vf-Chrimson and vf-Chrimson-ES/TS (Fig 1B). The ratio of membrane over intracellular immunofluorescence obtained from line profile analysis was significantly larger for vf-Chrimson-ES/TS (*P*-value = $1.33 \times 10^{-7}$) and f-Chrimson (*P*-value = $1.86 \times 10^{-7}$) than for vf-Chrimson (Fig 1 Bii), and similar to previously reported for Chronos-ES/TS (Keppeler *et al*, 2018). The expression of vf-Chrimson-ES/TS and vf-Chrimson was found across all turns of the injected cochlea (Fig 1C) with overall transduction rate of approximately 50% and a higher rate for apical SGNs in both cases (Fig 1D, apical vs. basal, *P*-value = 0.0014, Mann–Whitney *U*-test). We also found substantial expression in the contralateral, non-injected cochleae, indicating spread of virus in the specific conditions of pressure injection into the scala tympani of the early postnatal cochlea (Fig 1D). This spread likely occurred via the cochlear aqueduct and/or the endolymphatic ducts and the cerebrospinal fluid space (Lalwani *et al*, 1996). Clearly, incomplete and inhomogeneous SGN transduction along the tonotopic axis and viral spread indicates the need for further optimization of AAV-mediated gene transfer for improved efficiency and safety. The densities of SGNs in the injected and in non-injected ears were not significantly different from each other (Fig 1E). The injected mice behaved normally as concluded from routine animal observation and lacked obvious phenotypes such as circling, seizures, abnormal motor activity, or reduced body size.

oABRs were recorded in response to 1-ms-long 594 nm light pulses (1000 repetitions) delivered at 10 Hz via a 200-µm optical fiber that was inserted into the round window. oABRs were found in nine out of 11 AAV-PHP.eB-vf-Chrimson-ES/TS-injected mice and 13 out of 16 AAV-PHP.B-vf-Chrimson-injected mice (Fig 2A–C). Note that the negative deflection of the signal before stimulus onset results from low-pass filtering of the oABRs. oABRs of AAV-PHP.eB-vf-Chrimson-ES/TS-injected mice typically tended to show more differentiated waveforms, i.e., displaying more obvious ABR waves beyond wave I than those of AAV-PHP.B-vf-Chrimson-injected mice (Fig 2A). In both cases, the amplitude of oABR wave I, $P_1$-$N_1$, reflecting the optically evoked spiral ganglion compound action potential, grew with increasing stimulus intensity (Fig 2D) while its latency (i.e., the time interval between the stimulus onset and the oABR $P_1$ wave) got shorter (Fig 2E).

On average, the radiant flux at oABR threshold amounted to $6.88 \pm 1.72$ mW ($N = 9$ mice) for vf-Chrimson-ES/TS and

12.3 ± 2.81 mW (N = 13 mice) for vf-Chrimson (duration: 1 ms, rate: 10 Hz, 1000 repetitions, P-value = 0.1191, Mann–Whitney U-test). In most animals, the growth of oABR amplitudes did not saturate with increasing radiant flux, despite changes over more than an order of magnitude (Fig 2D). The minimal latency of the first oABR peak ($P_1$ latency at the maximum radiant flux; Fig 2E inset)

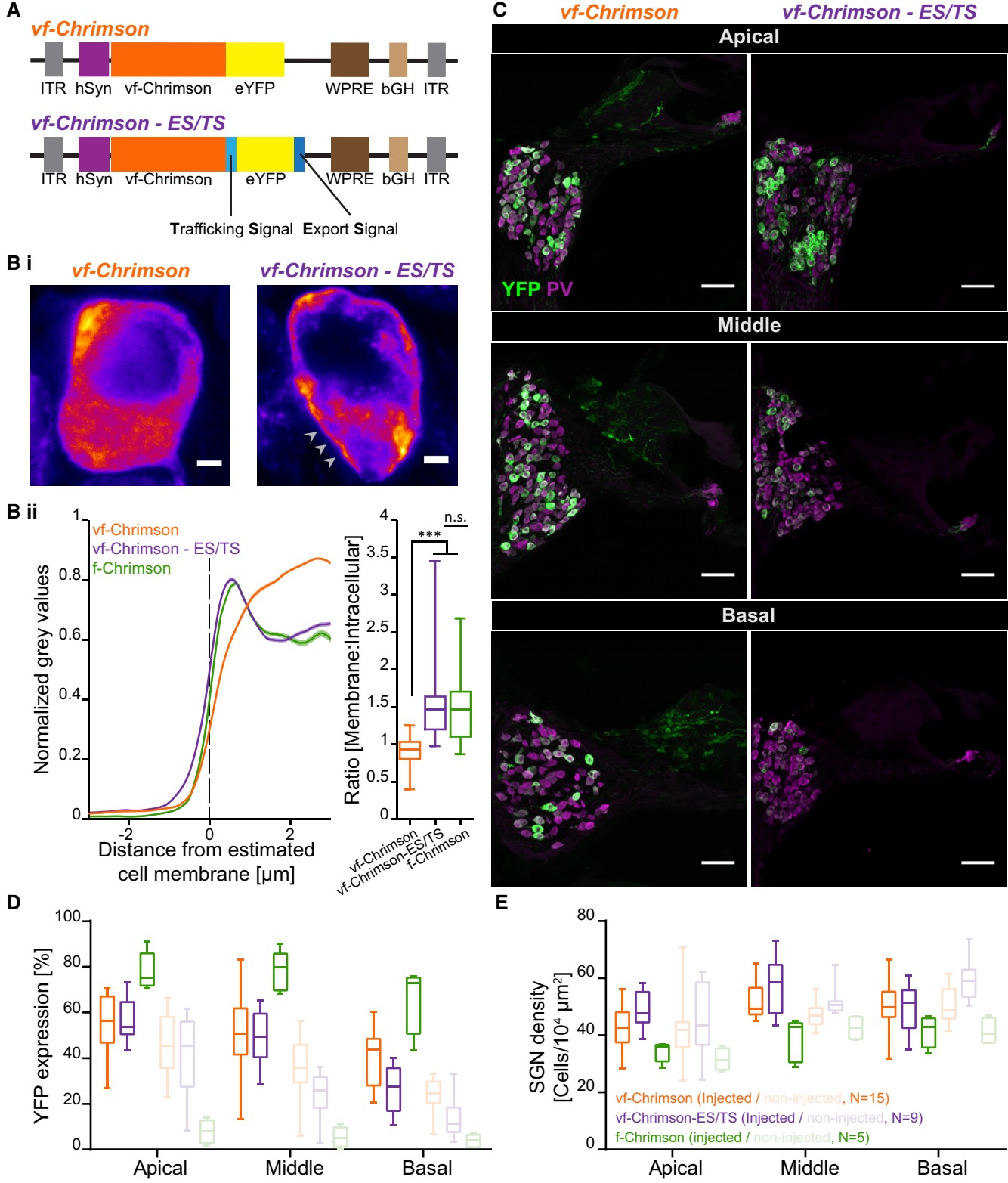

**Figure 1.**

**Figure 1. Establishing efficient expression of vf-Chrimson in SGNs.**

A  pAAV vector used in the study containing vf-Chrimson-eYFP ("vf-Chrimson", upper) or with a trafficking signal (TS), eYFP and ER export signal (ES) vf-Chrimson-eYFP ("vf-Chrimson-ES/TS", lower). In each, expression was driven by the human synapsin promoter (hSyn) and enhanced by the Woodchuck hepatitis virus posttranslational regulatory element (WPRE) and bovine growth hormone (bGH) polyadenylation signal (bGH poly A) sequences. ITR: inverted terminal repeats.

B  (Bi) Confocal images of eYFP-immunolabeled SGN somata transduced by either construct: Warmer colors represent higher fluorescence intensity indicating subcellular distribution of the channelrhodopsin expression. Enhanced localization of the opsin to the plasma membrane (arrowheads) is obvious when employing the ES/TS-trafficking signals. Scale bars: 2 μm. (Bii) Quantification of membrane localization of vf-Chrimson where fluorescence intensity of the immunofluorescence of the anti-YFP-antibody (mean ± SEM, line ± shaded area) is plotted across the cell membrane ($n = 30$ cells, $N = 5$ mice per group). The position of the plasma membrane was approximated where parvalbumin (PV) immunofluorescence of SGNs reached 50% (dashed line). For comparison, a f-Chrimson line profile was analyzed based on Mager et al (2018). Right panel: Box-and-whisker plots (minimum, 25th, median, 75th percentile, and maximum) of ratio of maximum membrane fluorescence to maximum intracellular fluorescence ($n = 30$ cells, $N = 5$ mice per group). vf-Chrimson scored around 1, which is lower compared to vf-Chrimson-ES/TS and f-Chrimson (f-Chrimson has intrinsically high membrane expression; $P$-value $= 1.1 \times 10^{-8}$, Kruskal–Wallis test followed by Tukey's multiple comparison test).

C  Representative maximum-projection confocal images of fluorescently labeled mid-modiolar cryosections of injected cochlea for vf-Chrimson and vf-Chrimson-ES/TS: PV-positive SGNs (magenta) and transduced SGNs (green) in apex, mid, and base of the cochlea. Scale bars= 50 μm.

D  Box-and-whisker plots (minimum, 25th, median, 75th percentile, and maximum) of the percentage of YFP-positive SGNs for all turns of injected (dark color) or contralateral non-injected (light color) cochleae. The horizontal line within the box indicates the median, boundaries of the box indicate the 0.25- and 0.75-percentile, and the whiskers indicate the highest and lowest values of the results. No significant differences are observed between vf-Chrimson and vf-Chrimson-ES/TS-injected ears ($N = 15$ mice for vf-Chrimson; $N = 9$ mice for vf-Chrimson-ES/TS; $P = 0.3596$, Mann–Whitney U-test).

E  Box-and-whisker plots (minimum, 25th, median, 75th percentile, and maximum) of the SGN density (number of PV-positive SGN somata per cross-sectional area of Rosenthal's canal) for all turns of injected (dark color) or contralateral non-injected (light color) cochleae. No significant differences were observed between vf-Chrimson and vf-Chrimson-ES/TS in the injected ear ($P$-value $= 0.1060$, Mann–Whitney test) nor between injected and non-injected cochleae of either construct (vf-Chrimson, $N = 15$ mice, $P$-value $= 0.2157$; vf-Chrimson-ES/TS, $N = 9$ mice, $P$-value $= 0.6517$, Mann–Whitney U-test).

Source data are available online for this figure.

was shorter for vf-Chrimson-ES/TS ($0.71 \pm 0.03$ ms, $N = 9$) than for vf-Chrimson-mediated oABRs ($1.05 \pm 0.05$ ms, $N = 13$, $P < 0.0001$, Mann–Whitney U-test). Next, we tested the dependence of oABRs on the stimulus rate (duration = 1 ms up to 500 Hz, duration = 0.5 ms from 500 Hz, maximum intensity: [38–43] mW, 1000 repetitions). When increasing stimulus rate, oABR amplitudes declined (Fig 2F) and latencies increased (Fig 2G). However, we could detect sizable $P_1$-$N_1$ (i.e., 20% of maximal oABR $P_1$-$N_1$ amplitude) up to stimulus rates of 500 Hz. oABR $P_1$-$N_1$ amplitudes were comparable between vf-Chrimson and vf-Chrimson-ES/TS for stimulation rates of 100, 300, and 500 Hz ($P > 0.05$, Kruskal–Wallis ANOVA, post hoc Dunn's multiple comparison test). For a reference, we replot the data from f-Chrimson-expressing SGNs recorded in our previous studies (Mager et al, 2018). Sizable oABRs could be elicited by light pulses shorter than 50 μs for both vf-Chrimson and vf-Chrimson-ES/TS (maximum intensity: 38–43 mW, rate: 10 Hz, 1,000 repetitions; Fig 2C). oABR amplitudes grew with pulse duration reaching the maximal $P_1$-$N_1$ amplitude earlier for vf-Chrimson-ES/TS (at approximately 0.4 ms) than vf-Chrimson (~ 0.8 ms, inset in Fig 2H, $P < 0.0001$, Mann–Whitney U-test). $P_1$-$N_1$ amplitudes tended to become smaller for longer pulses (and P1 latencies longer; Fig 2I), possibly due to accumulating channel inactivation and/or increasing depolarization block of SGNs upon prolonged photo-depolarization (Fig 2H). In summary, the oABR comparison indicates greater temporal fidelity but higher required radiant flux for both vf-Chimson- and vf-Chrimson-ES/TS-mediated SGN stimulation than previously found for f-Chrimson (Mager et al, 2018). This is in line with the shorter deactivation time constant and lower current density of vf-Chrimson expressed in NG cells (Mager et al, 2018).

**Ultrafast optogenetic stimulation of the auditory pathway: recordings from single putative SGNs**

To further evaluate the temporal fidelity of red-light optogenetic SGN stimulation mediated by vf-Chrimson-ES/TS and vf-Chrimson, we performed juxtacellular recordings from putative auditory nerve fibers (central axon of SGN) as previously described (Keppeler et al, 2018; Mager et al, 2018). As illustrated by the exemplary recordings (Fig 3A and B), putative SGNs fired upon optogenetic stimulation with high temporal precision for stimulus rates of up to few hundreds of Hz without an obvious advantage of vf-Chrimson-ES/TS over vf-Chrimson and adapted strongly for stimulation rate higher than 200 Hz. The temporal precision of firing, evaluated either as the vector strength (i.e., spike synchronization in one period of stimulation, Goldberg & Brown, 1969; Fig 4A) or as the spike jitter (i.e., standard deviation of spike latency across pulses; Fig 4B), was generally high up to 100 – 200 Hz of stimulation. Temporal precision and spike probability (Fig 4C) declined with increasing stimulation rates in both vf-Chrimson-ES/TS and vf-Chrimson-expressing putative SGNs in a similar manner. Spike jitter, calculated for spikes occurring in the time window between two pulse onsets, increased with stimulation rate but was typically below a millisecond for rates lower than 300 Hz (Fig 4B). At higher stimulation rates, spike jitter increased beyond the values obtained for simulated Poisson spike trains (see methods; Fig 4B), indicating spike synchronization with the light pulses became less reliable. We also calculated the spike latency (Fig 4D) within the same time window, which remained fairly stable up to 300 Hz. For a reference, we replotted the data from f-Chrimson-expressing putative SGNs recorded in our previous study (Mager et al, 2018). The comparison indicates comparable temporal fidelity of optogenetic SGN stimulation by vf-Chimson and f-Chrimson at least for stimulation rates up to 250 Hz. We did not find a major advantage of addition of the ES/TS sequences for vf-Chrimson-mediated optogenetic stimulation of single SGNs, at least when using AAV-PHP.eB as the viral vector, the human synapsin promoter and when AAV-injecting the cochlea at P6.

**Intensity coding by red ultrafast optogenetic stimulation**

Next, we studied the ability of putative SGNs to optogenetically encode changes in stimulus intensity. Single SGNs are able to

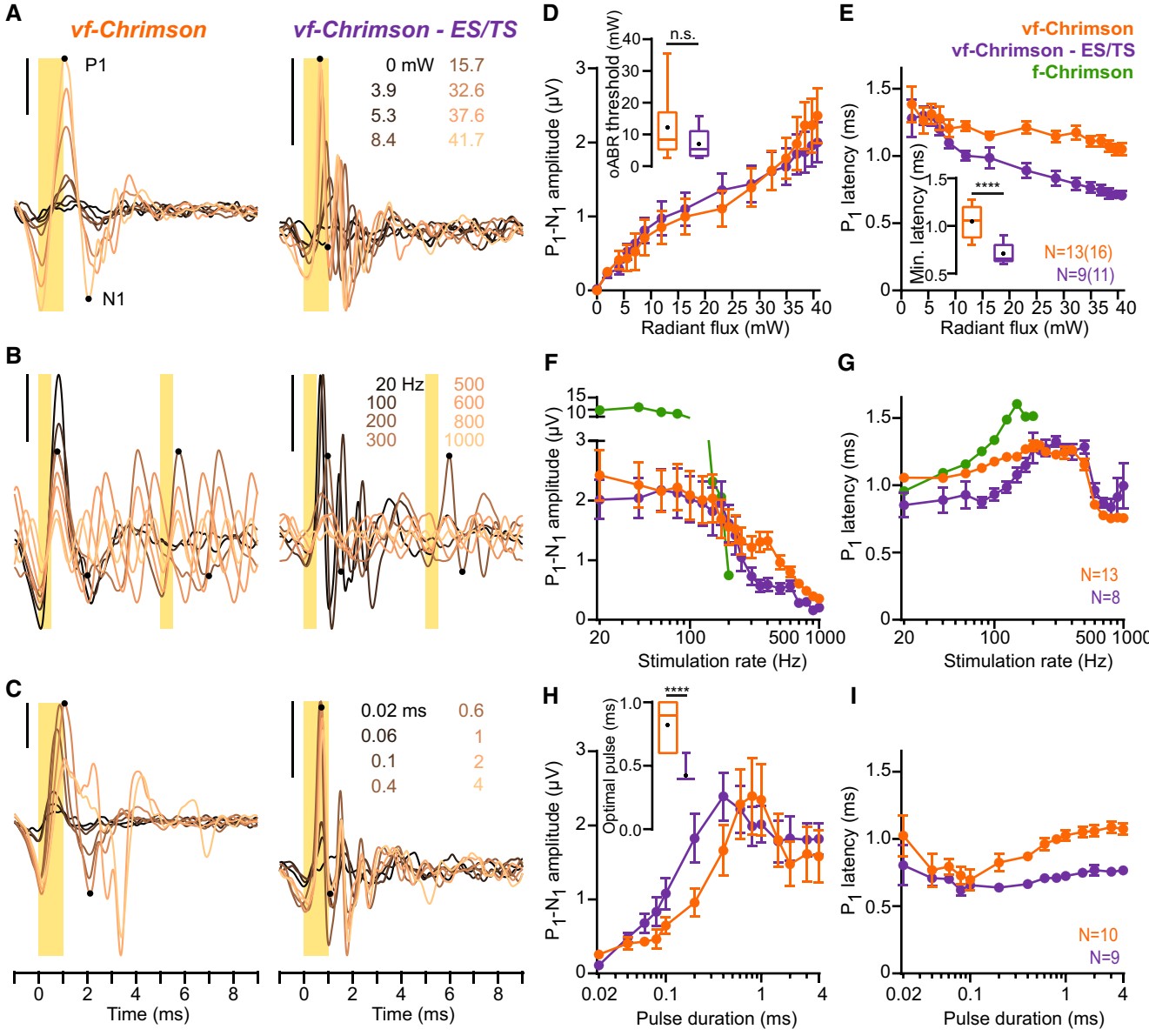

**Figure 2. Characterizing optogenetic stimulation of the auditory pathway by recordings of oABRs.**

A   Exemplary oABRs driven with varying radiant flux (1000 × 1 ms at 10 Hz, colors code the radiant flux in mW) of mice injected with AAV-PHP.B-vf-Chrimson (left) or AAV-PHP.B-vf-Chrimson-ES/TS (right). First positive ($P_1$) and first negative ($N_1$) peaks are indicated by black dots on the waveform triggered upon 41.7 mW radiant flux (the initial negative deflection results from filtering). Vertical scale bar: 1 μV.

B   Exemplary oABRs driven with varying stimulation rate (1 ms for data points ≤ 500 Hz and 0.5 ms above 500 Hz at 41.7 mW (highest radiant flux), colors code the stimulation rate) of the same animals from (A). Stimuli applied at a rate of 200 Hz and the corresponding $P_1$-$N_1$ pairs are indicated by yellow shaded area and black dots, respectively. Vertical scale bar: 1 μV.

C   oABRs driven with varying light pulse duration (10 Hz at 41.7 mW (highest radiant flux), colors code the pulse duration) of the same animals from (A). Exemplary pulse duration of 1 ms and corresponding $P_1$-$N_1$ pair are indicated by yellow shaded area and black dots. Vertical scale bar: 1 μV.

D–I   Quantification of $P_1$-$N_1$ amplitudes (mean ± SEM) and $P_1$ latencies (mean ± SEM) as a function of radiant flux (D, E, vf-Chrimson: $N$ = 13 mice, vf-Chrimson-ES/TS: $N$ = 9 mice), stimulation rate (F, G, vf-Chrimson: $N$ = 13 mice, vf-Chrimson-ES/TS: $N$ = 8 mice), and pulse duration (H, I, vf-Chrimson: $N$ = 10 mice, vf-Chrimson-ES/TS: $N$ = 9 mice). The average $P_1$-$N_1$ amplitude of f-Chrimson (green in F and G) is replotted from Mager *et al*, 2018. Inset in (D), quantification of the oABR threshold. Inset in (E), quantification of the shortest P1 latencies elicited among any radiant flux (****, $P$-value < 10⁻⁴, Mann–Whitney $U$-test). Inset in (H), quantification of the optimal pulse duration required to elicit the maximum $P_1$-$N_1$ amplitude. Boxes show 25th percentile, median, 75th percentile, the black dot the mean, and whiskers maximum and minimum.

Source data are available online for this figure.

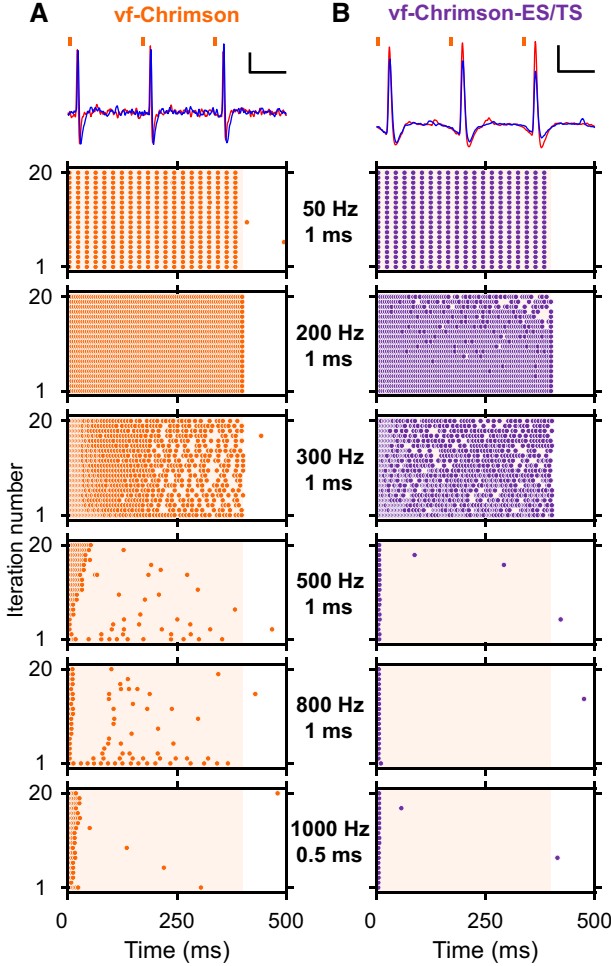

**Figure 3. Characterizing optogenetic temporal encoding by juxtacellular recordings from single putative SGNs.**

A, B  Exemplary spike traces at 50 Hz, 1 ms, maximum radiant flux and corresponding raster plots at varying stimulation rates of vf-Chrimson- (A, orange) and vf-Chrimson-ES/TS- (B, purple) expressing putative SGNs. Raster plots showing spiking activity of the above units in response to 400 ms-long trains of laser pulses (shaded areas, at 43 mW, pulse duration = 1 ms between 20 and 800 Hz, pulse duration = 500 μs for ≥ 900 Hz) recorded at six different stimulation rates over 20 iterations. Scale bars 0.1 mV, 10 ms (A) and 1 mV, 10 ms (B).

Bourien *et al*, 2014). Finally, dynamic range adaptation is thought to contribute to widen the range of intensities encoded by single SGN (Wen *et al*, 2009, 2012). In electrical hearing, the dynamic range at the single SGN level is very limited (~1 dB, Miller *et al*, 2006) and the extensive spread of excitation leads to massive recruitment of neighboring SGNs even for low to modest stimulus intensities. Hence, intensity coding by either mechanism is relatively poor with the eCI, which likely contributes to the limited speech perception in noise (Zeng & Galvin, 1999; Weiss *et al*, 2016).

How individual ChR-expressing SGNs respond to changes in light intensity has not yet been studied. However, en route to developing optogenetic hearing restoration, it is of great importance to probe over what range of light intensities SGN firing can be changed. Larger dynamic range of the response of single SGNs and higher spectral resolution would enable optogenetic SGN stimulation to improve loudness coding via both single SGN and SGN population activities. Here, we systematically compared intensity coding between the acoustic and optogenetic SGN stimulation using f-Chrimson, vf-Chrimson, and vf-Chrimson-ES/TS. We first compared the responses of naïve putative SGNs to acoustic click trains (from 20 to 300 Hz, 400 ms duration, 100 ms recovery, 20 iterations, 300 μs width, 100 dB SPL [peak equivalent, pe]) to the response of f-Chrimson expressing putative SGNs to trains of light pulses (1-ms pulse duration, 18.3 mW radiant flux; Fig EV2). Up to 100 Hz, the firing probability was similar between the acoustic and optogenetic stimulation (Wilcoxon rank-sum test). At higher stimulation rate, the firing probability declined as a function of stimulation frequency, which was more pronounced for the optogenetic stimulation (200 Hz, $P$-value = $1.5 \times 10^{-4}$; 300 Hz, $P$-value = $8.2 \times 10^{-5}$, Wilcoxon rank-sum test). As the firing probability were comparable between the two stimulation modalities up to a stimulation rate of 100 Hz, this rate was chosen to evaluate SGN responses to changing stimulus intensity.

Single putative SGNs of naïve and optogenetically modified cochleae were recorded in response to 350- or 400-ms trains of acoustic or optical stimulation followed by 150- or 100-ms recovery (1 trial = 500 ms, 20 trials, 300-μs acoustic clicks, or 1-ms light pulses) at as many intensities that could be applied while recording a given SGNs (Fig 5A). A total number of 77 putatives SGNs from 12 mice were included for subsequent analysis: 19 acoustic SGNs ($N$ = 3 mice), 14 f-Chrimson SGNs ($N$ = 2 mice), 26 vf-Chrimson SGNs ($N$ = 2 mice), and 37 vf-Chrimson-ES/TS SGNs ($N$ = 5 mice). Figure 5B shows the relation between firing rate and stimulus intensity (rate-level function, RLF).

Prior to the recordings from single putative SGNs, the a/oABRs were recorded upon acoustic or optogenetic stimulation for different intensities in order to determine the threshold of the SGN population response (Table 1). The sound or light intensity for ABR threshold was then used as reference in order to present the response of single putative SGNs as a function of stimulus intensity in $dB_{ABR\ thr}$ in a given animal (Fig 5C; Materials and Methods). This facilitated the comparison of intensity encoding between both modalities and reduced the impact of inter-animal variance. Next, we determined the rate-based threshold of each putative SGNs using detection theory (d' statistics, similar to level yielding to 10% of maximum driving rate, $P$-value = 0.96, Macmillan & Creelman, 2004; Huet *et al*, 2018) with the silence/dark condition as the reference (Fig 5C, Table 1, Huet *et al*, 2018) which was later used to align growth function across SGNs (Figs 6 and 7).

encode changes in sound intensity of about 5–50 dB SPL as measured in response to pure tones (Sachs & Abbas, 1974; Liberman, 1978; Winter *et al*, 1990; Taberner & Liberman, 2005; Huet *et al*, 2016). However, the full range of sound intensities audible is much larger, e.g., spanning six orders of magnitude or 120 dB in humans (Viemeister & Bacon, 1988). This discrepancy is known as the "dynamic range problem" (Evans, 1981). To achieve wide dynamic range coding at the SGN population level, at least to some degree, the individual SGNs of one tonotopic place contribute in a complementary way to cover the full range of audible sound pressure level. Furthermore, in natural hearing, increments in sound pressure level correlate with increased proportion of recruited SGNs which also contribute to loudness coding (Furman *et al*, 2013;

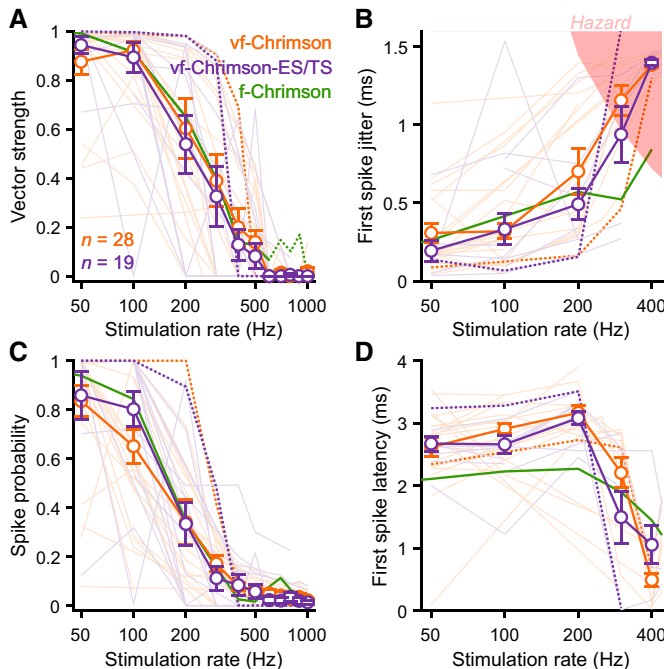

**Figure 4. Analysis of temporal fidelity of vf-Chrimson-mediated optogenetic SGN stimulation.**

A–D Quantification of the vector strength (A), first spike jitter (B), spike probability (C), and first spike latency (D) as a function of the repetition rate of putative SGNs expressing vf-Chrimson (orange, $n = 28$ putative SGNs) or vf-Chrimson-ES/TS (purple, $n = 19$ putative SGNs). Single SGNs are represented in light and mean ± SEM per vf-Chrimson variant in color. f-Chrimson data (green) were replotted from Mager *et al* (2018). The putative SGNs represented on Fig 3 are shown with dashed lines. In (B), the red-shaded area represents the hazard function (i.e., the averaged first spike latency jitter measured from simulated Poisson spike train not containing any synchronization).

Source data are available online for this figure.

For both stimulation modalities, firing rates increased with stimulus intensity. For clicks, most RLFs grew with sound intensity showing "sloping saturation" (i.e., firing rate increased with sound intensity but did not saturate completely; Fig 6A,E,I, Sachs & Abbas, 1974). In contrast, firing of most light-stimulated SGNs showed "flat saturation" (Fig 6B–D,F–H,I, Sachs & Abbas, 1974). Further quantifications of the intensity coding were assisted by fitting a sigmoidal function to the RLF (Fig 5C) allowing to determine the saturation level, the dynamic range (i.e., level difference to threshold yielding a driven rate change equal to 90% of the maximum driving rate; Ohlemiller *et al*, 1991) and the RLF slope (i.e., the ratio between 90% of the maximum driving rate and the dynamic range). RLF growth was typically monotonic for both stimulation modalities, but we mention that 33% of f-Chrimson and 13.33% of vf-Chrimson expressing putative SGNs had a monotonicity index ≤ 0.9 (Table 1). At saturating stimulation level, across modalities and opsin variants, most SGNs fired ~ 1 spikes per light pulse (Table 1). The first spike (FS) evoked per stimulus occurred ~0.85 ms later for acoustic clicks than for the two vf-Chrimson variants (vf-Chrimson: *P*-value = 0.02; vf-Chrimson-ES/TS: *P*-value = $1 \times 10^{-4}$, Kruskal–Wallis test followed by a multi-comparison test). Unexpectedly, the FS latency

of f-Chrimson expressing SGNs was also significantly longer than for vf-Chrimson-ES/TS (*P*-value = $7.9 \times 10^{-3}$) and previously reported data (Mager *et al*, 2018).

Next, we quantified the adaptation ratio (i.e., the ratio of discharge rate during the first 100 ms and 350 ms) at the threshold, at the mid-intensity eliciting 50% of the maximal firing rate and at saturating intensity (Fig 6J). For the three optogenetic variants, the firing was phasic (adaptation ratio > 1) at threshold and mid-intensity but tonic at saturating intensity (adaptation ratio = 1; Fig 6A–D, J, table 1). In response to acoustic click, in contrast, the firing was tonic from threshold to mid-intensity. We then quantified the dynamic range (Figs 5C and 6E–H,K) and also compared the averaged RLF per group aligned to the SGN threshold (Fig 6I). The dynamic range tended to be smaller for optogenetic than for acoustic stimulation (Fig 6E–H,K, Table 1), which reached statistical significance for vf-Chrimson and vf-Chrimson-ES/TS SGNs (vf-Chrimson: *P*-value = $8.66 \times 10^{-6}$; vf-Chrimson-ES/TS: $7.83 \times 10^{-4}$). Surprisingly, the dynamic range of vf-Chrimson-mediated firing was also smaller than that found with f-Chrimson (vf-Chrimson: *P*-value = $8 \times 10^{-4}$; vf-Chrimson-ES/TS: $1.14 \times 10^{-2}$, Kruskal–Wallis test followed by a multi-comparison test). The RLF slope was steeper for optogenetic stimulation (Fig 6L, Table 1). Considering the pronounced adaptation of optogenetically evoked firing, we also quantified the dynamic range and RLF slope for the first 100 ms of stimulation, which, however, did not yield a different outcome (Fig 6K and L).

As mentioned above, SGNs with the diverse RLFs are thought to complement each other providing the brain with information over a broader range of stimulus intensities than covered by the individual SGN. To estimate the operating range of the population of acoustically and optogenetically driven SGNs, we first applied a sigmoidal fit to the average discharge rate (Fig 6E–H) and extracted the population dynamic range which amounted to 43.9 dB (pe SPL) for acoustic and 15.1, 5.9, and 7.11 dB (mW) for f-Chrimson, vf-Chrimson, and vf-Chrimson-ES/TS-mediated optogenetic stimulation. We then averaged the RLFs of the three most and least sensitive SGNs and calculated the difference in laser power between them at 10% (most sensitive) and 90% (least sensitive) of activation which amounted to 55.22 dB (pe SPL) and 11.51, 2.96, and 11.66 dB (mW) for f-Chrimson, vf-Chrimson, and vf-Chrimson-ES/TS-mediated optogenetic stimulation. We repeated the analysis of most and least sensitive putative SGNs for individual mice and also observed a wider dynamic range than for the individual optogenetically driven SGNs: f-Chrimson: 10.64 dB ($n = 6$), vf-Chrimson: 2.96 dB ($n = 15$), vf-Chrimson-ES/TS: 5.77 dB ($n = 13$). This suggests that the heterogeneous responses to optogenetic stimulation found at the single SGN level widen the dynamic range of stimulus intensity encoding at the population level.

### Temporal properties of SGN coding as a function of stimulus intensity

Next, we analyzed the temporal firing properties of the same putative SGNs for different stimulus intensities. As illustrated in Fig 5D, the FSL declined with increasing stimulus intensity (Fig 7A–D). Note that for reasons currently not fully understood, FSL in this new f-Chrimson data (Fig 7B) was longer than previously found (Mager *et al*, 2018; replotted in Fig 4D). At saturating stimulus intensity, the

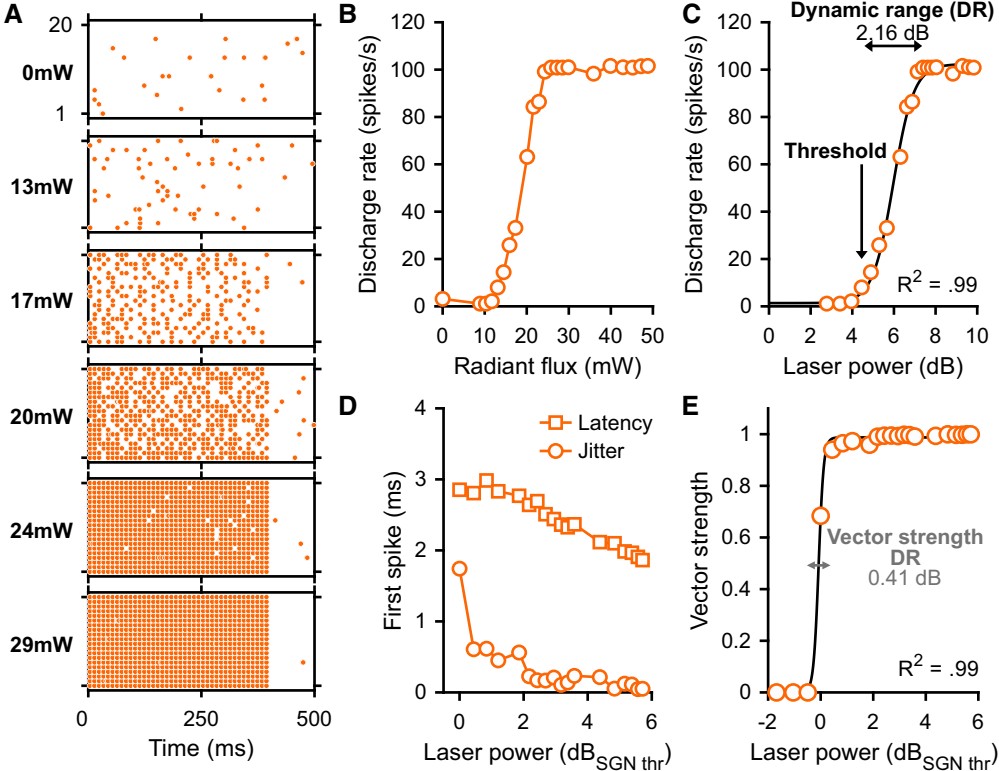

**Figure 5. Intensity encoding analysis of a putative SGN expressing vf-Chrimson.**

The spike train analysis was performed on the first 350 ms following the onset of the light pulse train. Temporal fidelity parameters (e.g., vector strength, first spike latency, and first spike jitter) were computed on the first spike evoked by each light pulse.

A   Raster plots from a vf-Chrimson-expressing putative SGN at a subset (0, 13, 17 20, 24 29 mW) of the different tested radiant fluxes.

B   Discharge rate as a function from the radiant flux from the SGN presented in (A).

C   Rate-level function (RLF) of the SGN presented in (A). The laser power was calculated as follows: Laser power (in dB [mW]) = $10 \times \log_{10}\left(\frac{A}{A_0}\right)$ where A is the radiant flux (in mW) and $A_0$ is the oABR threshold (in mW). The RLF was fitted using a sigmoidal fit (black line, $R^2 = 0.99$) and the dynamic range, displayed by a double arrow, defined as the level difference yielding a driven rate change equal to 90% of the maximum driving rate (Ohlemiller *et al*, 1991). The threshold was determined as the lowest laser power for which a d' of 1, using the dark condition as reference and was computed from the discharge rate (Macmillan & Creelman, 2004; Huet *et al*, 2018).

D   First spike latency (FSL, square) and first spike jitter (FSJ, circle) level functions.

E   Vector strength-level function (VS-LF) centered on the rate-based threshold. The VS-LF was fitted using a sigmoidal fit and the dynamic range, displayed by the black line above the VS-LF, defined as the level difference yielding a driven VS change equal to 90% of the maximum driving VS.

mean FSL was significantly shorter for vf-Chrimson and vf-Chrimson-ES/TS-mediated optogenetic stimulation than in response to clicks (Table 1, vf-Chrimson: *P*-value = $9 \times 10^{-4}$, vf-Chrimson-ES/TS: *P*-value = $6.93 \times 10^{-8}$, Kruskal–Wallis test followed by a multi-comparison test).

The temporal precision of firing was analyzed by means of the vector strength (VS, Fig 5E) and first spike jitter (Fig 5D). At maximum stimulus intensities, spike jitter tended to be lower, i.e., more precise spike timing, for optogenetic stimulation (Fig 7E–H, Table 1). The jitter of the first spike was significantly smaller for both vf-Chrimson variants than for acoustic clicks (vf-Chrimson: *P*-value = $1.6 \times 10^{-3}$, vf-Chrimson-ES/TS: *P*-value = $9.67 \times 10^{-7}$) and for f-Chrimson (*P*-value = $4.2 \times 10^{-3}$ for comparison to vf-Chrimson-ES/TS, Fig 7E–H). For both optogenetic and acoustic stimulation (100 Hz trains as above), VS increased steeply with stimulus intensity around the spike rate threshold and reached the maximum possible VS of 1, indicating intensity-dependent synchronization of

the spikes with the stimulus (Fig 7I–L, Table 1). Once maximum VS was reached, it tended to decay at higher intensities for acoustic stimulation, while it remained high for optogenetic stimulation. We quantified the range of intensities over which VS changes (VS dynamic range) by sigmoidal fitting of the VS-level function (insert in Fig 7I–L, Table 1). The VS dynamic range tended to be smaller for optogenetic than for acoustic stimulation, which reached significance for vf-Chrimson and vf-Chrimson-ES/TS (vf-Chrimson: *P*-value = $5 \times 10^{-4}$; vf-Chrimson-ES/TS: $1.88 \times 10^{-5}$, Kruskal–Wallis test followed by a multi-comparison test).

Finally, we quantified the covariation of the temporal fidelity measures and the discharge rate (Fig 8). For all modalities, VS saturated with intensity prior to the discharge rate (Fig 8A–D). For acoustic stimulation, first spike latency, its jitter and discharge rate seemed to saturate for similar click intensities (Fig 8E and I). In contrast, for optogenetic stimulation, first spike latency and its jitter continued to decrease at intensities for which the discharge rate was

**Table 1.  Quantification of intensity encoding by putative SGNs.**

| | Acoustic click | f-Chrimson | vf-Chrimson | vf-Chrimson-ES/TS |
|---|---|---|---|---|
| ABR threshold in dB (pe SPL) or dB (mW)[a] | 42.5 ± 2.16 | 1 ± 00 | 5.85 ± 0.81 | 8.61 ± 2.22 |
| Activation threshold (dB rel. to ABR threshold: $dB_{ABRthr}$)[b] | 29.74 ± 0.71 | 2.60 ± 0.32 | 8.92 ± 0.05 | 3.73 ± 0.10 |
| Monotonicity index[b] | 0.99 ± 0.00 | 0.93 ± 0.01 | 0.97 ± 0.01 | 0.99 ± 0.00 |
| Saturation level (dB rel. to ABR threshold: $dB_{ABRthr}$)[b] | 54.00 ± 1.38 | 7.47 ± 0.33 | 9.89 ± 0.3 | 4.49 ± 0.14 |
| Spike probability at saturating level (spikes/s)[b] | 1.24 ± 0.04 | 1.06 ± 0.02 | 0.92 ± 0.02 | 1.01 ± 0.02 |
| First spike latency at the saturating level (ms)[b] | 2.93 ± 0.04 | 2.77 ± 0.07 | 2.19 ± 0.03 | 1.99 ± 0.02 |
| Adaptation ratio at threshold[b] | 0.94 ± 0.01 | 2.43 ± 0.07 | 1.84 ± 0.04 | 1.82 ± 0.03 |
| Adaptation ratio at mid-intensity[b] | 1.02 ± 0.01 | 1.18 ± 0.02 | 1.19 ± 0.01 | 1.18 ± 0.01 |
| Adaptation ratio at saturating intensity[b] | 1.07 ± 0.01 | 1.02 ± 0.01 | 1.11 ± 0.02 | 1.00 ± 0.00 |
| Dynamic range (dB)[b] | 24.61 ± 1.61 | 3.77 ± 0.13 | 0.67 ± 0.04 | 0.97 ± 0.03 |
| Slope (spikes × s$^{-1}$/dB)[b] | 5.61 ± 0.30 | 26.77 ± 1.07 | 259.98 ± 15.12 | 169.86 ± 8.61 |
| First spike latency at maximum intensity (ms)[b] | 2.93 ± 0.03 | 2.36 ± 0.04 | 2.17 ± 0.02 | 1.82 ± 0.02 |
| Maximum vector strength (VS)[b] | 0.936 ± 0.004 | 0.993 ± 0.000 | 0.991 ± 0.000 | 0.996 ± 0.000 |
| dynamic range of VS (dB)[b] | 4.16 ± 0.27 | 1.63 ± 0.23 | 0.435 ± 0.02 | 0.79 ± 0.06 |
| First spike jitter at maximum intensity (ms)[b] | 1.02 ± 0.04 | 0.27 ± 0.01 | 0.13 ± 0.00 | 0.11 ± 0.00 |

Average ± SEM of the different quantified variables of the intensity encoding for the acoustic and optogenetic (Chrimson, vf-Chrimson, and vf-Chrimson-ES/TS) stimulation.
[a]Acoustic click, *N* = 3 mice; f-Chrimson, *N* = 2 mice; vf-Chrimson, *N* = 2 mice; vf-Chrimson-ES/TS, *N* = 5 mice.
[b]Acoustic click, *n* = 19 putatives SGNs; f-Chrimson, *n* = 14 putatives SGNs; vf-Chrimson, *n* = 26 putatives SGNs; vf-Chrimson-ES/TS, *n* = 37. Intensities were expressed as dB (pe SPL) for clicks and dB (mW) for light pulses, see Materials and Methods.

already saturated (Fig 8F–H,J–L). This suggests a broader dynamic range based on the temporal fidelity than on the discharge rate for optogenetic stimulation.

## Discussion

Here, we characterized the optogenetic stimulation of SGNs mediated by the ultrafast red-light-activated ChRs f-Chrimson and vf-Chrimson. Following early postnatal injection of AAVs into the mouse cochlea, both ChRs were expressed very well by SGNs. Adding trafficking sequences derived from K$^+$ channel Kir2.1 to vf-Chrimson improved plasma membrane expression, yet little benefit was found for optogenetic SGN stimulation contrasting previous observations with the ultrafast blue-light-activated ChR Chronos. Neural population responses of vf-Chrimson-expressing SGNs showed amplitudes greater than 200 nV (~10% of maximal oABR amplitude) for stimulation rates ≥ 500 Hz. On average, spike probability and phase locking quality of single putative SGNs expressing vf-Chrimson were near 100% at 100 Hz and dropped to 50% around 200 Hz of stimulation for strong light pulses. This temporal fidelity of vf-Chrimson-mediated SGN stimulation is comparable to that of f-Chrimson- and Chronos-ES/TS-expressing SGNs. The intensity dependence of the SGN firing rate showed a smaller dynamic range for optogenetic than for acoustic stimulation. It spanned 3.77, 0.67, and 0.97 dB (mW) for f-Chrimson, vf-Chrimson, and vf-Chrimson-ES/TS, respectively, for individual SGNs, and rose to 11.51, 2.96, and 11.66 at the level of the recorded SGN population. In contrast to acoustic stimulation, the temporal fidelity of optogenetically triggered SGN firing (first spike latency and first spike jitter) did not saturate. This suggests that spike timing lends to optogenetic

intensity encoding over a broader range of light intensity than firing rate. In conclusion, vf-Chrimson and f-Chrimson are valuable candidates for red-light optogenetic SGN stimulation.

### Ultrafast vf-Chrimson-mediated stimulation of the auditory pathway

Stimulation of the auditory pathway is a prime example for an application of optogenetics requiring both high temporal fidelity and light sensitivity. Upon sound stimulation in physiological hearing, SGNs fire at hundreds of Hz and show sub-millisecond temporal precision of spiking relative to the stimulus. Electrical stimulation achieves even higher temporal precision (Miller *et al*, 2006) such that eCIs typically employ very high stimulation rates (800 Hz and greater) to avoid overly synchronized activity in the auditory nerve that otherwise would cause an unnatural hearing percept (Zeng, 2017). Here, we probed the utility of ultrafast ChR with red-shifted action spectrum for stimulating SGNs firing. To a first approximation, one would expect the deactivation (closing) kinetics of the expressed ChR and radiant flux to govern the temporal fidelity of SGN firing. Consistent with this notion, we find that the faster the closing kinetics of the ChR, the higher the stimulation rate that the SGN population can respond to with a sizable compound action potential (20% of maximal oABR $P_1$-$N_1$ amplitude): 200 Hz for f-Chrimson with $\tau_{off}$: 3 ms (Mager *et al*, 2018), 500 Hz for vf-Chrimson with $\tau_{off}$: 1.6 ms (Mager *et al*, 2018), and 1 kHz for Chronos with $\tau_{off}$: 0.7 ms (Keppeler *et al*, 2018). At the level of single putative SGNs, on average, 50% spike probability and vector strength of 0.5 were observed at approximately 200 Hz of stimulation for f-Chrimson (Mager *et al*, 2018), vf-Chrimson (this study), and Chronos-ES/TS (Keppeler *et al*, 2018). We note that some putative SGNs showed good responses to

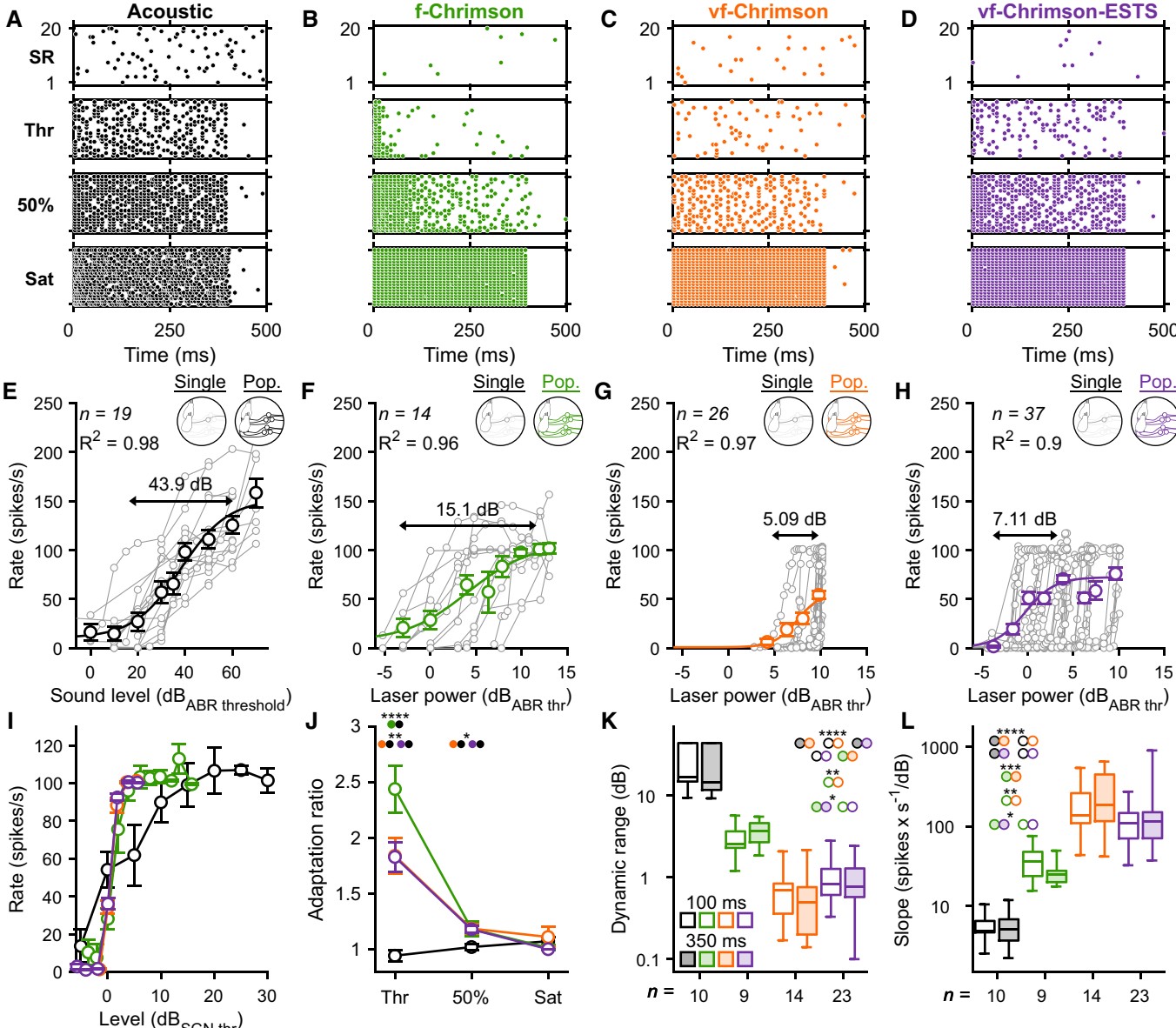

**Figure 6. Stimulus intensity encoding upon acoustic and optogenetic stimulation of putative SGNs.**

Recordings were made using 350 or 400 ms click/light pulse train followed by 150 or 100 ms silence/dark (1 trial = 500 ms). The discharge rate was measured from the first 350 ms following the beginning of the click/light pulse train. Only putative SGNs for which spontaneous rate (SR) was recorded were included as follows: SR was used to determine the threshold using d' statistics (Macmillan & Creelman, 2004; Huet *et al*, 2018). Rate-level function (RLF) of putative SGNs, from which evoked-firing was recorded for at least 4 stimulus intensities between threshold and saturation, was fitted with a sigmoid function to determine their slope, mid-intensity (eliciting 50% of maximum driving rate), saturation, and dynamic range (i.e., level difference yielding a driven rate change equal to 90% of the maximum driving rate). Kruskal–Wallis test followed by a multi-comparison test (*: $P$-value $\leq 5 \times 10^{-2}$,**: $P$-value $\leq 10^{-2}$, ***: $P$-value $\leq 10^{-3}$, ****: $P$-value $\leq 10^{-4}$).

A–D Representative raster plots from acoustically (A, 300 µs acoustic click) and optogenetically (B, f-Chrimson; C, vf-Chrimson; D, vf-Chrimson-ES/TS; 1 ms light pulse, λ = 594 nm) stimulated putative SGNs at different intensities: no click/light (SR), threshold (Thr), 50% and saturation (Sat).

E–H RLFs for acoustic (E, $n = 19$ SGNs, $N = 3$ mice) and optogenetic (F, f-Chrimson, $n = 14$ SGNs, $N = 2$ mice; G, vf-Chrimson, $n = 26$ SGNs, $N = 2$ mice; H, vf-Chrimson-ES/TS, $n = 37$ SGNs, $N = 5$ mice) SGN responses. Single RLFs are represented in gray; population RLFs were binned (bin width = 2 dB) and represented as average ± SEM. Population RLFs were fitted by a sigmoid function in order to extract the population dynamic range (reported in black above the population RLFs). The goodness of fit was expressed as $R^2$.

I Averaged threshold-aligned acoustic and optogenetic RLFs (bin width= 2 dB, mean ± SEM) from the SGNs presented in (E-H).

J Adaptation ratio (i.e., ratio between discharge rate measured from the first 100 ms and 350 ms) at threshold, mid-intensity (50%), and saturated driven rate quantified from the SGNs presented in (E-H), average ± SEM.

K, L Quantification of the acoustic and optogenetic RLFs dynamic range (J) and slope (K), same SGNs presented in (E-H), computed from the first 100 ms (white fill) and 350 ms (colored fill) of the responses to click/light pulse trains. Boxes show 25th percentile, median, 75th percentile, and whiskers maximum and minimum.

Source data are available online for this figure.

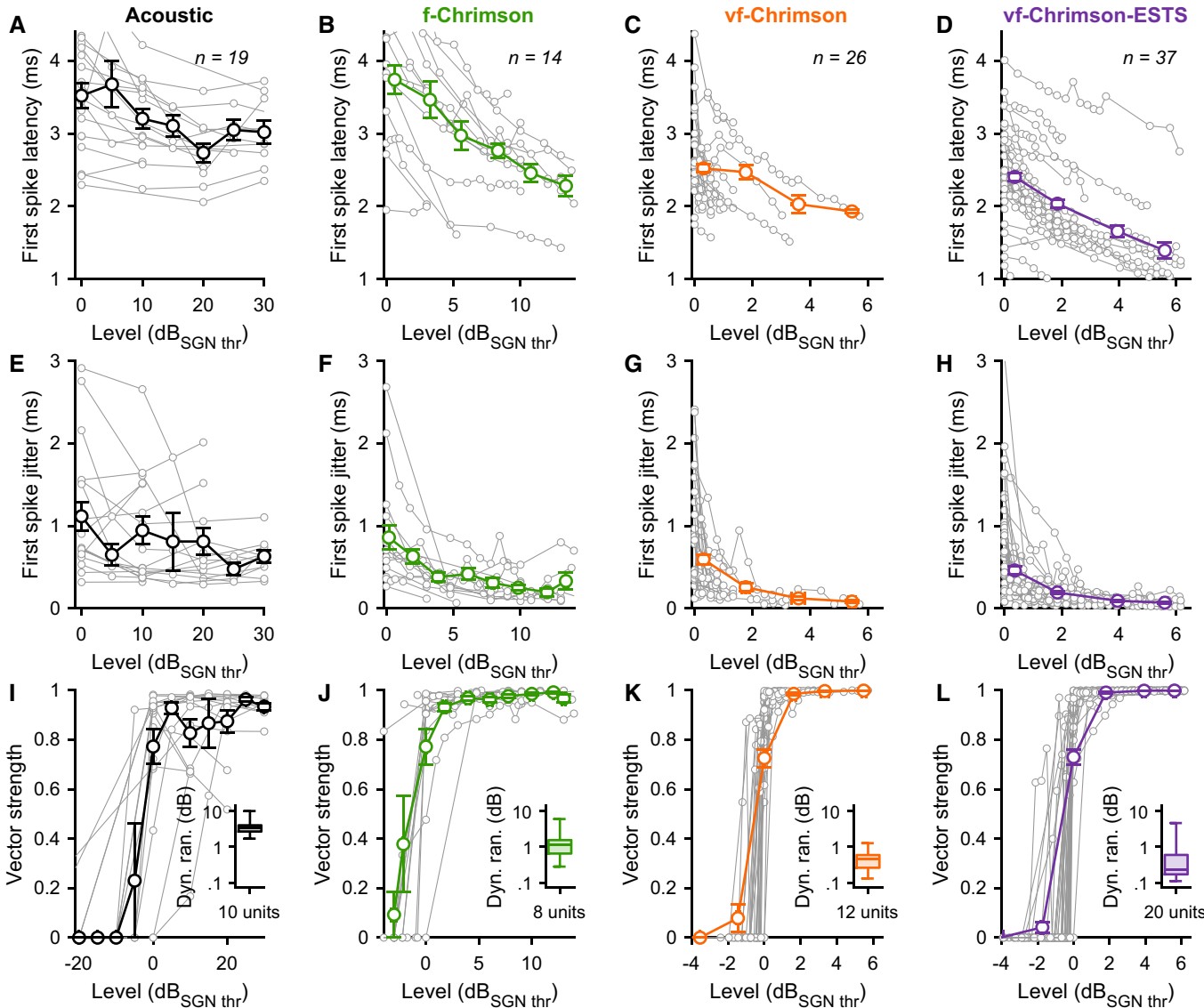

**Figure 7. Temporal fidelity of firing as a function of the stimulus intensity: first spike latency (FSL, A-D) and jitter (FSJ, E-H) and vector strength (VS, I-L).**

A–L  VS, FSL, and FSJ were computed from the first spike elicited by each click/light pulse. Single VS-, FSL- and FSJ-level functions, from the putative SGNs presented in Fig 5, were aligned on the rate-based threshold and plotted in gray. Averaged (acoustic: bin = 5 dB; f-Chrimson/vf-Chrimson/vf-Chrimson-ES/TS: bin = 2 dB) level functions were plotted in color using the same color code than in Fig 5. Average ± SEM. Inserts in (I-L) : Quantification of the dynamic range (10 – 90% of the difference between SR and saturation) quantified from the single VS-level functions using a sigmoid fit. Boxes show 25th percentile, median, 75th percentile, and whiskers maximum and minimum. For acoustic, n = 19 SGNs, N = 3 mice; for f-Chrimson, n = 14 SGNs, N = 2 mice; for vf-Chrimson, n = 26 SGNs, N = 2 mice; for vf-Chrimson-ES/TS, n = 37 SGNs, N = 5 mice.

Source data are available online for this figure.

even higher rates of stimulation (e.g., Fig 4 of this study and Keppeler *et al*, 2018).

Higher temporal fidelity of optogenetic coding comes at a price. The faster channel closing, i.e., the shorter the open channel lifetime, the less depolarizing charge is contributed by the individual open ChR upon stimulation by brief light pulses. Therefore, shorter open channel lifetime results in higher thresholds for optical stimulation: The radiant flux threshold for oABR was found to be 6.6 mW for Chronos-ES/TS (Keppeler *et al*, 2018), 6.9 mW for vf-Chrimson-ES/TS (this

study), and 0.5 mW for f-Chrimson (Mager *et al*, 2018). Membrane expression of the ChR co-determines the light sensitivity conveyed to neurons. This likely explains at least part greater light sensitivity conveyed by f-Chrimson to SGNs than by vf-Chrimson, as f-Chrimson showed greater photocurrent densities at saturating irradiance also in NG cells. Efforts to optimize the membrane trafficking and residence of ChRs are critical for reducing the required light doses and the proteostatic stress of SGNs. These preclinical efforts should also aim for using trafficking sequences of human membrane

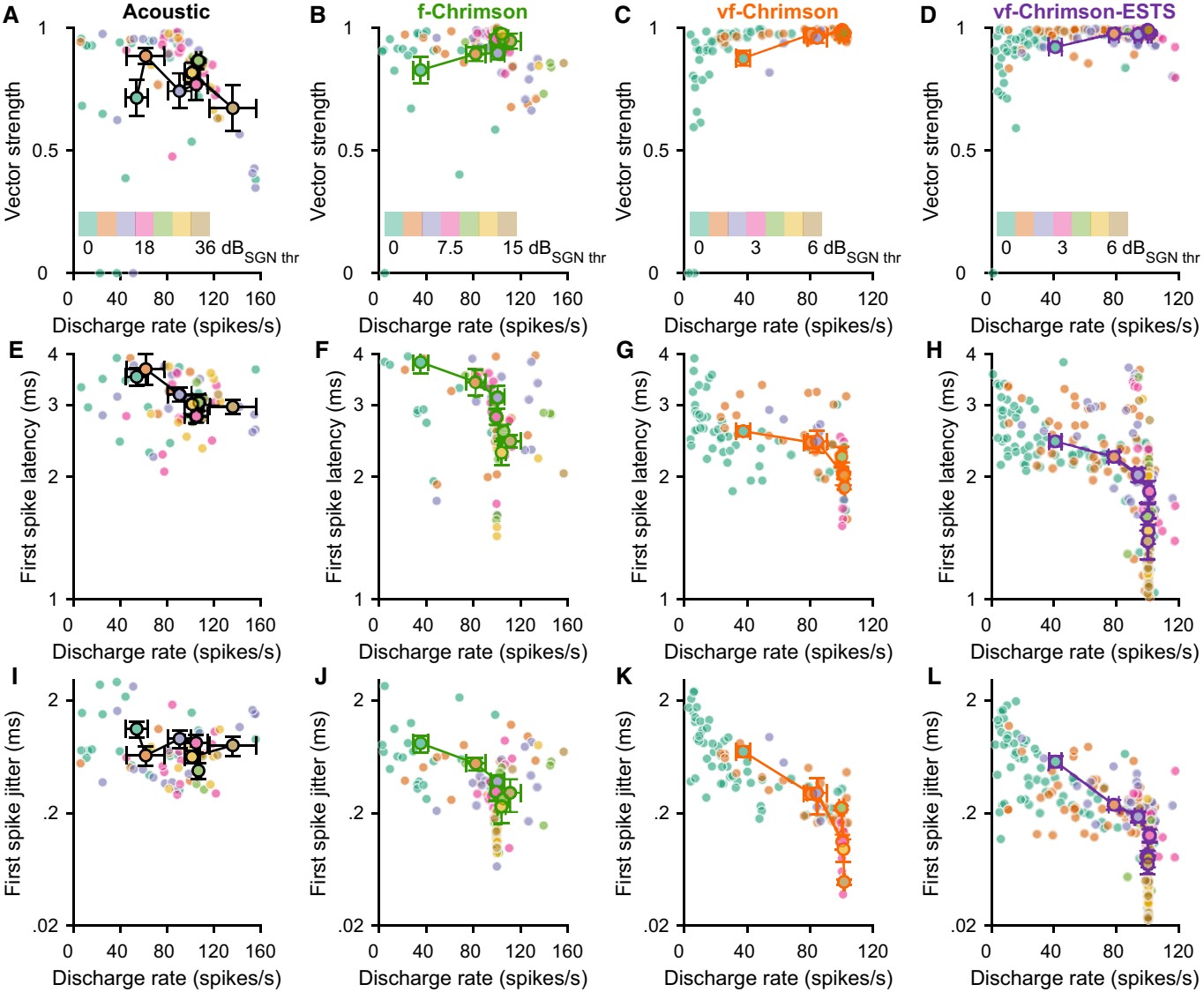

**Figure 8. Comparing intensity dependence of discharge rate and temporal fidelity of firing.**

A–L  Vector strength- (A-D), FSL-(E-H), and FSJ- (I-L) as a function of the discharge rate. The intensity above threshold (dB SL) is encoded by the color scale presented in (A-D, acoustic: bin width = 6 dB; f-Chrimson: bin width = 2.5 dB; vf-Chrimson/vf-Chrimson-ES/TS: bin width = 1 dB). Mean ± SEM were plotted using as marker edge color, the color corresponding to the group and the intensity above threshold is indicated by the marker face color. For acoustic, *n* = 19 SGNs, *N* = 3 mice; for f-Chrimson, *n* = 14 SGNs, *N* = 2 mice; for vf-Chrimson, *n* = 26 SGNs, *N* = 2 mice; for vf-Chrimson-ES/TS, *n* = 37 SGNs, *N* = 5 mice.

Source data are available online for this figure.

proteins (such as Kir2.1 in the present study) and evaluate the effect of omitting the fluorescent protein from the expression construct to enhance biosafety (preprint: Gauvain *et al*, 2020). While we found significantly shorter oABR latency and significantly lower optimal pulse duration, as well as a tendency for lower oABR thresholds for the trafficking-optimized vf-Chrimson-ES/TS, the optogenetic stimulation remained inferior in amplitude to that mediated by f-Chrimson (e.g., Fig 2). This might at least in part relate to the lower transduction rate observed in the present study upon early postnatal cochlear AAV injection despite the use of likely more potent viral vectors (AAV-PHP.B and AAV-PHP.eB with

transduction rates being statistically indistinguishable from each other) than used for f-Chrimson here and in Mager *et al* (2018, AAV6) with the same means of administration. More generally, we note that differences in the viral vector and its titer as well as trial to trial variability of the AAV injection might contribute to the variance in the functional responses such as oABR.

The utility of optogenetic stimulation for temporal encoding by SGNs in comparison with acoustic stimulation was addressed using f-Chrimson, vf-Chrimson, and vf-Chrimson-ES/TS. Up to 100 Hz, we found significantly higher temporal precision of firing for optogenetic than for acoustic stimulation. When driving SGNs at saturating

stimulus intensities, the temporal jitter was lower and the vector strength was slightly higher for optogenetic stimulation (Figs 7 and 8, Table 1). While interpretation of the latter needs to consider the lower spontaneous firing rate in the case of optogenetic experiments, we regard the lower jitter of evoked spikes to primarily reflect the bypassing of the stochastic processes of sensory transduction and synaptic transmission in physiological coding. Temporal precision of coding increased with the level of optogenetic stimulation while it saturated acoustically. Interestingly, the vector strength and spike probabilities approached 1 already for low stimulus levels in the case of optogenetic stimulation and remained high for higher stimulation levels. In contrast, for click stimulation vector strength tended to grow more gradually with spike probability and to decline with the highest levels. In conclusion, optogenetic stimulation with f-Chrimson and vf-Chrimson-ES/TS provides near-physiological temporal fidelity of coding.

### Intensity coding SGNs expressing fast Chrimson

Next to the utility of optogenetic stimulation for coding temporal (see above and (Keppeler *et al*, 2018; Mager *et al*, 2018) and spectral (Hernandez *et al*, 2014; Dieter *et al*, 2019) information, the range of stimulus intensities over which the firing rate of SGNs change is an important parameter to study. So far estimates of this so-called output dynamic range of the activation of the auditory pathway were based either on oABR amplitude (approximately 10–16 dB [mW], Mager *et al*, 2018; Wrobel *et al*, 2018) or multi-unit firing rate in the inferior colliculus (approximately 11 dB [mW], Dieter *et al*, 2019). It is important to note here that in all cases, saturation of the response was not reached with the radiant fluxes used; hence, these estimates represent an "apparent dynamic range" as a lower bound. Importantly, the range over which the individual SGN change their firing probability had not yet been investigated. Here, using f-Chrimson, vf-Chrimson, and vf-Chrimson-ES/TS-mediated fiber-based optogenetic stimulation we found the output dynamic range at the single SGN level to range between 0.67 and 3.77 dB (mW). This presents an advantage over the output dynamic range of electrical SGN stimulation ( ~1 dB [current level], Miller *et al*, 2006) but still does not get close to our average estimate of SGN stimulation by acoustic clicks (~25 dB [pe SPL]). When deriving a population operating range over which spike rate changes from the sample of recorded f-Chrimson, vf-Chrimson, vf-Chrimson-ESTS-expressing SGNs, we obtained dynamic ranges between 2.96 and 11.51 dB (mW). The apparently greater dynamic range obtained for f-Chrimson expressing putative SGNs compared with vf-Chrimson variant requires careful consideration. In fact, our f-Chrimson putative SGN data set exhibited longer first spike latency than previously reported data (Mager *et al*, 2018) and vf-Chrimson variants (this study) expressing SGNS. Therefore, it cannot be excluded than a part of the sample contained ventral cochlear nucleus neurons which: *i*) acoustically exhibits bigger dynamic range than auditory nerve fibers (Rhode & Smith, 1986) and *ii*) is not distinguishable from SGN when optogenetically driven (Keppeler *et al*, 2018; Mager *et al*, 2018). Finally, the temporal optogenetic response (first spike latency and first spike jitter) did not saturate at high stimulation intensities, suggesting that if integrated by the auditory brainstem neurons, it could serve encoding of a larger dynamic range than reported in this study. Obviously, the lifetime of an SGN recording did not allow us to grade the stimulus intensity in a very fine-grained manner as would be useful to derive physiological estimates of discernible intensity levels. This calls for behavioral experiments to provide intensity discrimination limens (King *et al*, 2016).

### Toward developing the optogenetic cochlear implant: evaluating the utility of optogenetic SGN stimulation in comparison with physiological and electrical stimulation

This study used recordings of population and single SGN activity to parametrize the utility of optogenetic SGN stimulation for coding time and intensity information. The obtained estimates provide important input for planning sound coding strategies for future optical CIs. It becomes clear that coding strategies should target stimulation rates < 500 Hz for efficient yet sufficiently stochastic coding, rendering unnecessary the very high stimulation rates employed in eCIs. Indeed, studies of speech understanding as a function of stimulation rate in eCI users report no further gain beyond 500 Hz (Shannon *et al*, 2011). In order to maintain a reasonable battery lifetime, lowering the stimulation rate from the current state of art seems imperative: as the energy requirement per pulse of optogenetic stimulation (several μJ) currently exceeds that of the electrical cochlear implants (less than a μJ, Zierhofer *et al*, 1995), and since the goal is to significantly increase the number of stimulation channels given the greater frequency selectivity (Dieter *et al*, 2019). Clearly, the light emitted from simultaneously activated stimulating channels will partially overlap, and hence, the energy budget will likely scale sublinearly with the number of channels in the oCI. Nonetheless, from the energetic point of view it seems that the apparent dynamic range found for optogenetic stimulation will be a realistic estimate also for a clinical implementation.

## Materials and Methods

### Preparation of expression constructs

We used pcDNA-vf-Chrimson-eYFP plasmid (generous gift from Ernst Bamberg) as starting material the preparation of expression constructs. This construct was digested with BamHI/HindIII (NEB) restriction enzymes, and the fragment containing vf-Chrimson-eYFP was gel-extracted (Zymo Research) and further used for ligation. At the same time, the plasmid pAAV_hSyn_f-Chrimson-eYFP (Mager *et al*, 2018) was also digested using restriction enzymes BamHI/HindIII and used as a backbone plasmid. The human Synapsin promoter was used to drive transgenic expression of opsins in SGNs. This material was used to perform In-fusion cloning (TaKaRa/Clontech) and a PCR with the following primers: 5'-AATTCAAGCTGC TAGCATGGCTGAGCTGATCAG-3' and 5'-CCTGCTCTTGACCGGTC ACTGTGTCCTCGT-3'. In the second step, the obtained PCR fragment was gel extracted (Zymo Research) and used for In-fusion ligation with the backbone plasmid pAAV_hSyn_Chronos-ES/TS (Keppeler *et al*, 2018) derived from digestion with the restriction enzymes NheI/AgeI (NEB). All obtained ligation products were further tested by the restriction enzyme digestion and finally sequenced externally.

## Virus purification

AAVs were generated in HEK-293T cells (ATCC) using polyethylen-imine transfection (25,000 MW, Polysciences, USA, Gray *et al,* 2011; Deverman *et al,* 2016). The cell line was regularly tested negatively for mycoplasma. In brief, triple transfection of HEK-293T cells was performed using the pHelper plasmid (TaKaRa/Clontech), the trans-plasmid providing viral capsid PHP.B (generous gift from Ben Deverman and Viviana Gradinaru, Caltech, USA), or PHP.eB (PHP.eB was a gift from Viviana Gradinaru (Addgene plasmid # 103005; http://n2t.net/addgene:103005; RRID:Addgene_103005) and the cis-plasmid providing vf-Chrimson or vf-Chrimson-ES/TS (Fig 1A). We harvested viral particles 72 h after transfection from the medium and 120 h after transfection from cells and the medium. Viral particles from the medium were precipitated with 40% polyethylene glycol 8000 (Acros Organics, Germany) in 500 mM NaCl for 2 h at 4°C and, after centrifugation at 4,000 g for 30 min, combined with cell pellets for processing. The cell pellets were suspended in 500 mM NaCl, 40 mM Tris, 2.5 mM MgCl2, pH 8, and 100 U/ml of salt-activated nuclease (Arcticzymes, USA) at 37°C for 30 min. Afterward, the cell lysates were centrifuged at 2,000 *g* for 10 min and AAVs purified over iodixanol (OptiPrep, Axis Shield, Norway) step gradients (15, 25, 40, and 60%, Zolotukhin *et al,* 1999; Grieger *et al,* 2006) at 58,400 rpm for 2.25 h. AAVs were concentrated using Amicon filters (EMD, UFC910024) and formulated in sterile phosphate-buffered saline (PBS) supplemented with 0.001% Pluronic F-68 (Gibco, Germany). Virus titers were measured using an AAV titration kit (TaKaRa/Clontech) according to manufacturer's instructions by determining the number of DNase I resistant vg using qPCR (StepOne, Applied Biosystems). Purity of produced viruses was routinely checked by silver staining (Pierce, Germany) after gel electrophoresis (Novex™ 4–12% Tris-Glycine, Thermo Fisher Scientific) according to manufacturer's instruction. The presence of viral capsid proteins was positively confirmed in all virus preparations. Viral stocks were kept at −80°C until the injection.

## NG108-15 cell culture and transfection

NG108-15 cells (ATCC, HB-12377TM, Manassas, USA) were cultured at 37°C and 5% CO2 in DMEM (Sigma, St. Louis, USA) supplemented with 10% fetal calf serum (Sigma, St. Louis, USA) and 5% penicillin/streptomycin (Sigma, St. Louis, USA). Transient transfections with pcDNA3.1(-) derivatives carrying f-Chrimson-EYFP and vf-Chrimson-EYFP using Lipofectamine LTX (Invitrogen, Carlsbad, USA) were performed two to three days prior to the confocal live cell imaging experiments.

## Postnatal AAV injection into the cochlea

Postnatal AAV injection into scala tympani of the left ear via the round window was performed at p6 wild-type C57BL/6 mice essentially as described in Huet and Rankovic (2021) using AAV2/6, AAV-PHP.B, or AAV-PHP.eB suspensions. In brief, under general isoflurane anesthesia and local analgesia achieved by means of xylocaine, the left ear was approached via a dorsal incision and the cochlea position estimated through the cartilaginous bulla. A borosilicate capillary pipette containing the virus was insert in the

cochlea and kept in place to inject approximately 1–1.5 μl of AAV2/6_hSyn-Chrimson ($9.9 \times 10^{12}$ genome copies/ml), PHP.B_hSyn-vf-Chrimson-eYFP ($8.7 \times 10^{12}$ genome copies/ml), or PHP.eB_hSyn-vf-Chrimson-ES/TS ($1.1 \times 10^{13}$ genome copies/ml). After virus application, the tissue above the injection site was repositioned, the wound sutured and buprenorphine (0.1 mg kg$^{-1}$) was applied as a pain reliever. Recovery of the animals was then tracked daily. In all experiments, mice were randomly selected for injection. Hence, surgery prior to stimulation needed to be done in the injected ear. Animals were then kept in a 12-h light/dark cycle, with access to food and water *ad libitum*. All experiments were done in compliance with the national animal care guidelines and were approved by the board for animal welfare of the University Medical Center Göttingen and the animal welfare office of the state of Lower Saxony (LAVES; 14/1726 and 17/2394).

## Immunostaining and imaging of cochlear cryosections

Cochleae were fixed with 4% paraformaldehyde in phosphate-buffered saline (1 h). Sections of the cochlea were cryosectioned following 0.12 M EDTA decalcification. After incubation of sections for 1 h in goat serum dilution buffer (16% normal goat serum, 450 mM NaCl, 0.6% Triton X-100, 20 mM phosphate buffer, pH 7.4), primary antibodies were applied over night at 4°C. The following antibodies were used as follows: chicken anti-GFP (catalog no.: ab13970, Abcam, 1:500) and guinea pig anti-parvalbumin (catalog no.: 195004, Synaptic Systems, 1:300). Thereafter, secondary AlexaFluor-labeled antibodies (goat-anti-chicken 488 IgG (H + L), catalog no.: A-11039, Thermo Fisher Scientific, 1:200; goat-anti guinea pig 568 IgG (H + L), catalog no. A1107, Thermo Fisher Scientific, 1:200) were applied for 1 h at room temperature. Confocal images were collected using a SP5 microscope (Leica) and processed in ImageJ. Expression was considered positive when anti-GFP immunofluorescence in a given cell (marked by anti-parvalbumin immunofluorescence) was found to be higher than 3xSD above the background fluorescence.

For analysis of ChR distribution, line profiles (length: 7.5 μm, width: 3 pixels) were aligned to the NG (visually aligned on the cytosol) and SGN (approximated as the position for which parvalbumin immunofluorescence rose to 50% of its intracellular value) cell membrane. The line profiles were oriented perpendicular to the cell edge. For membrane/intracellular expression ratio, a maximum peak detection was performed for membranous area (defined as 0 μm) and for intracellular area (defined as 1.12 μm).

## Optical stimulation in vivo

The procedure was done under general isoflurane anesthesia, and analgesia was achieved by means buprenorphine (0.1 mg/kg) and local administration of xylocaine. Body temperature was maintained at 37°C using a custom-designed heat plate. The left (AAV-injected) middle ear was reached using a retroauricular approach and opened to expose the cochlea. A 50- or 200-μm optical fiber coupled to a 594 nm laser (OBIS LS OPSL, 100 mW, Coherent Inc.) was inserted into the cochlea via the round window. Irradiance was calibrated with a laser power meter (LaserCheck; Coherent Inc.).

## Auditory brainstem responses

For stimulus generation and presentation, data acquisition, and offline analysis, we used a NI System and custom-written MATLAB software (The MathWorks, Inc.). Optically evoked ABRs (oABRs) and acoustically evoked ABRs (aABRs) were recorded by needle electrodes underneath the pinna, on the vertex, and on the back near the legs. The difference potential between vertex and mastoid subdermal needles was amplified using a custom-designed amplifier, sampled at a rate of 50 kHz for 20 ms, filtered (300–3,000 Hz), and averaged across 1000 presentations. The first ABR wave was detected semi-automatically with a custom-written MATLAB script in which the wave was detected for each trace in a temporal window defined by the user. Thresholds were determined by visual inspection as the minimum sound or light intensity that elicited a reproducible response waveform in the recorded traces.

## Juxtacellular recordings from single putative SGNs

Mice were anesthetized with an intraperitoneal injection of xylazine (5 mg/kg) and urethane (1.32 mg/kg), and analgesia was achieved with buprenorphine (0.1 mg/kg). Body temperature was maintained at 37°C using a custom-designed heat plate, placed on a vibration isolation table in a sound-proof chamber (IAC GmbH, Niederkrüchten, Germany). A tracheostomy was performed prior to positioning the animals in a custom-designed stereotactic head holder. After securing the position, pinnae were removed, scalp reflected, portions of the lateral interparietal and of the left occipital bone removed, to allow for a partial aspiration of the cerebellum and expose the surface of the cochlear nucleus. Glass microelectrodes (~50 MΩ) were advanced through the posterior end of the anteroventral cochlear nucleus using an LN Mini 55 micromanipulator (Luigs & Neumann, Germany) and aimed toward the internal auditory canal. Action potentials were amplified using an ELC-03XS amplifier (NPI Electronic, Tamm, Germany), filtered (300–20000 Hz), digitized (National Instruments card PCIe-6323), analyzed, and prepared for display using custom-written MATLAB (The MathWorks, Inc.) software. For the temporal encoding experiment, when light-responsive fibers were found, 400 ms-long trains of 1 ms-long pulse presented at repetition rates between 20 and 1,000 Hz were presented, leaving 100 ms inter-train recovery over 20 iterations for each tested rate. Different rates were tested following no particular order, except that 20 Hz was the first repetition rate presented across all units. For repetition rates higher or equal to 200 Hz, parameters were computed if the spike probability was equal to or greater than 5%. Otherwise, values were set to 0 for spike probability and discharge rate and to "not a number" for first spike latency, jitter, and vector strength. Phase locking was quantified using the vector strength (Goldberg & Brown, 1969), considering a cycle starting at the onset of a light pulse and ending at the onset of the subsequent pulse, and conforming to the equation:

$$\text{vector strength} = \frac{\sqrt{\left[\sum_{i=1}^{n}\cos\theta_i\right]^2 + \left[\sum_{i=1}^{n}\sin\theta_i\right]^2}}{n}, \Theta_1, \Theta_2, \ldots, \Theta_n \text{ cycle phases}$$

in which spikes occurred. The Rayleigh test was used to evaluate the significance of vector strength: if L > 13.8, the null hypothesis is rejected at the 0.001 significance level (Hillery & Narins, 1987) and insignificant VS was set to 0. The spike probability was calculated as the ratio between the number of spikes and the number of light pulses. The temporal jitter is the standard deviation of spike latency across trials measured in one period of stimulation. The hazard function (for the temporal jitter analysis) was calculated for each stimulation rate by simulating spiking as a Poisson process at given rates (from 10 to 1000 spikes/s).

For the intensity encoding experiment, 350/400 ms-long click (300 μs) / light pulse (1 ms) trains at a repetition rate of 100 Hz were presented, leaving 150/100 ms inter-train recovery over 20 iterations for each tested intensity. Different intensities were tested at no particular order, except that maximum laser output and no stimulation were the first tested intensities across all units. Spike train analysis was performed on the first 350 ms of the evoked response. Discharge rate and spike probability were computed as described above. First spike latency, jitter, and vector strength were computed as described on the first spike evoked by each acoustic click/light pulse. The monotonicity index was calculated as the ratio between the firing rate at maximum intensity and the maximal firing rate. In order to directly compare intensity encoding for acoustic and optogenetic stimulation and for reducing effects of variance across recordings from different mice, for Fig 5 C we related stimulus intensity to that eliciting ABR threshold. Sound intensity in $dB_{ABRthreshold}$ (pe SPL) = $20 \times \log_{10}(A/A_0)$ where A is the presented sound pressure and $A_0$ = sound pressure at aABR threshold. Optical intensity in $dB_{ABR\ threshold}$ (mW) = $10 \times \log_{10}(A/A_0)$ where A is the presented radiant flux and $A_0$ the radiant flux at oABR threshold. Rate and vector strength-level function were fitted using a sigmoidal fit if at least four stimulus intensities were falling between threshold and saturation of the growth function. If the coefficient of determination ($R^2$) was greater than 0.8, the sigmoidal fit was used to extract the level at 50% activation, the saturation level, the dynamic range (i.e., level difference yielding a driven rate change equal to 90% of the maximum driving rate), and the slope. The rate-based threshold was determinate using a d' statistical test with the silent/dark condition as reference (Macmillan & Creelman, 2004; Huet et al, 2018). The d' threshold was similar to the intensity yielding to 10% of sigmoidal fit (P-value = 0.96).

## Data analysis

The data were analyzed using MATLAB (MathWorks), Excel (Microsoft), FIJI (ImageJ2), Origin (Microcal Software), and GraphPad Prism (GraphPad Software). In Fig 8, the ColorBrewer color map was used (Cobeldick, 2018). Averages were expressed as mean ± SEM or mean ± SD, as specified in the captions. References to data in the main text were expressed as mean ± SEM. For statistical comparison between two groups, data sets were tested for normal distribution (the D'Agostino & Pearson omnibus normality test or the Shapiro–Wilk test or Jarque–Bera test) and equality of variances (F-test) followed by two-tailed unpaired Student's t-test, or the unpaired Wilcoxon rank-sum test when data were not normally distributed and/or variance was unequal between samples.

For evaluation of multiple groups, statistical significance was calculated by using one-way ANOVA test (equality of variances tested with the Brown–Forsythe test) or one-way Kruskal–Wallis test followed Tukey's honest significant difference criterion test.

### The paper explained

#### Problem

With its 700,000 users, the electrical cochlear implant is currently the reference approach to enable hearing in deaf people. Yet, its ability to restore speech comprehension in noisy background or music appreciation is limited. This results from the broad spread of electric current from each electrode contact that activates large sets of spiral ganglion neurons (SGNs) spanning a broad range of frequencies. Future optogenetic cochlear implants aim to overcome this bottleneck by harnessing the utility of focused light for spatially selective activation of SGNs. Here, the quest lies at finding the most suitable channelrhodopsin (ChR) to render SGNs light sensitive. The ideal light-gated channel should (i) allow reliable and safe optogenetic modification of SGNs; (ii) be activated by long-wavelength light to avoid phototoxicity; (iii) require low light intensity to open; and (iv) close fast to allow fast temporal encoding.

#### Results

Here, fast (f-) and very fast (vf-) variants of Chrimson, a red-shifted ChR, were evaluated for their utility in encoding time and intensity information in mouse SGNs. vf-Chrimson enabled SGNs to fire at near-physiological rates with good temporal precision up to 250 Hz of stimulation. The dynamic range of optogenetic stimulation was narrower than for acoustic clicks but larger than that reported for electrical stimulation. Comparative investigation of optogenetic stimulation by f- and vf-Chrimson versus acoustic clicks with respect to spike rate encoding of light intensity suggests that f-Chrimson can accommodate a broader dynamic range than vf-Chrimson at lower light costs.

#### Impact

Optogenetics offers an expanding range of natural and engineered opsins for neuronal control. Yet, only few provide fast switching and confer high light sensitivity at a longer wavelength. Our results characterize two of these opsins, f- and vf-Chrimson, as candidates for use in future clinical and for addressing research questions especially in the field of auditory neuroscience.

## Data availability

This study includes no data deposited in external repositories.

**Expanded View** for this article is available online.

## Acknowledgements

We thank Daniela Gerke for expert help with virus and immunolabeled mid-modiolar cochlear cryosection preparation, Christiane Senger-Freitag and Sandra Gerke for expert technical support, Gerhard Hoch for his expert engineering support, and Patricia Räke-Kügler for excellent administrative support. We thank Ben Deverman and Viviana Gradinaru for providing the PHP.B construct used in this study. This work was funded by the European Research Council through the Advanced Grant 'OptoHear" to TM under the European Union's Horizon 2020 Research and Innovation program (grant agreement No. 670759), the Fraunhofer and Max-Planck cooperation program (NeurOpto grant) to TM and was further supported by the German Research Foundation through the Cluster of Excellence (EXC2067) Multiscale Bioimaging to TMa, AH, and TM as well as the Leibniz Program to TM, and a scholarship of the Göttingen Promotionskolleg für Medizinstudierende, funded by the Jacob-Henle-Programm or Else-Kröner-Fresenius-Stiftung (Promotionskolleg für Epigenomik und Genomdynamik, 2017_Promotionskolleg.04) to AM. In addition, this research is supported by Fondation Pour l'Audition (FPA RD-2020-10) to TM. Open Access funding enabledand organized by Projekt DEAL.

## Author contributions

TM, AH, VR, BB, and DLM designed the study. BB, DLM, and AM performed o/aABR and single SGNs recordings. BB, TMa, and AH performed immunohistochemistry and FP expression analysis. VR performed preparation of expression construct, virus purification and production, and injections of the viruses. BB, AH, and DLM analyzed the data and prepared the figures. TM, AH, and VR supervised the work. All authors contributed to analyze of the data and contributed to the writing of the manuscript.

## Conflict of interest

TM is a co-founder and CEO of OptoGenTech company. The other authors declare no conflict of interests.

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
