## [Review Process File · EMBO Molecular Medicine]

Utility of red-light ultrafast optogenetic stimulation of the auditory pathway

Burak Bali, David Lopez de la Morena, Artur Mittring, Thomas Mager, Vladan Rankovic, Antoine Huet, and Tobias Moser

DOI: [10.15252/emmm.202013391](https://doi.org/10.15252/emmm.202013391)

Corresponding authors: Tobias Moser (tmoser@gwdg.de) , Antoine Huet (antoine.huet@med.uni-goettingen.de), Vladan Rankovic (vrankovic@dpz.eu)

Review Timeline:

Submission Date:	4th Sep 20
Editorial Decision:	30th Sep 20
Revision Received:	2nd Feb 21
Editorial Decision:	26th Feb 21
Revision Received:	22nd Mar 21
Accepted:	23rd Mar 21

Editor: Jingyi Hou

Transaction Report:

30th Sep 2020

Dear Prof. Moser,

Thank you again for the submission of your manuscript to EMBO Molecular Medicine. We have now received feedback from the three referees whom we asked to evaluate your manuscript. As you will see from the reports below, the referees acknowledge the interest and the high technical quality of the study. However, they also raise a series of concerns about your work, which need to be convincingly addressed in a major revision of the present manuscript.

Without repeating all the points raised in the reviews below, some of the most substantial issues are the following:

- additional quantitative experiments regarding gene delivery and expression are required to improve the conclusiveness of the study, as suggested by Referee #1.
- the link between the presented findings and future clinical applications needs to be better discussed.
- the clarity in data/study presentation needs to be improved to make the findings easily accessible to the general audience of EMBO Molecular Medicine.

All other issues raised by the referees need to be satisfactorily addressed as well. We would welcome the submission of a revised version within three months for further consideration. Please note that EMBO Molecular Medicine strongly supports a single round of revision and that, as acceptance or rejection of the manuscript will depend on another round of review, your responses should be as complete as possible.

We are aware that many laboratories cannot function at full efficiency during the current COVID-19/SARS-CoV-2 pandemic and have therefore extended our "scooping protection policy" to cover the period required for a full revision to address the experimental issues. Please let me know should you need additional time, and also if you see a paper with related content published elsewhere.

I look forward to receiving your revised manuscript.

Sincerely,
Jingyi

Jingyi Hou

*** Instructions to submit your revised manuscript ***

**** PLEASE NOTE **** As part of the EMBO Publications transparent editorial process initiative (see our Editorial at <https://www.embopress.org/doi/pdf/10.1002/emmm.201000094>), EMBO Molecular Medicine will publish online a Review Process File to accompany accepted manuscripts.

To submit your manuscript, please follow this link:

Link Not Available

- 1) a .docx formatted version of the manuscript text (including Figure legends and tables). Please make sure that the changes are highlighted to be clearly visible to referees and editors alike.
- 2) separate figure files*
- 3) supplemental information as Expanded View and/or Appendix. Please carefully check the authors guidelines for formatting Expanded view and Appendix figures and tables at <https://www.embopress.org/page/journal/17574684/authorguide#expandedview>
- 4) a letter INCLUDING the reviewers' reports and your detailed responses to their comments (as Word file)

Also, and to save some time should your paper be accepted, please read below for additional information regarding some features of our research articles:

- 5) The paper explained: EMBO Molecular Medicine articles are accompanied by a summary of the articles to emphasize the major findings in the paper and their medical implications for the non-specialist reader. Please provide a draft summary of your article highlighting
 - the medical issue you are addressing,
 - the results obtained and
 - their clinical impact.

This may be edited to ensure that readers understand the significance and context of the research.

Please refer to any of our published articles for an example.

6) For more information: There is space at the end of each article to list relevant web links for further consultation by our readers. Could you identify some relevant ones and provide such information as well? Some examples are patient associations, relevant databases, OMIM/proteins/genes links, author's websites, etc...

7) Author contributions: the contribution of every author must be detailed in a separate section (before the acknowledgments).

8) EMBO Molecular Medicine now requires a complete author checklist (<https://www.embopress.org/page/journal/17574684/authorguide>) to be submitted with all revised manuscripts. Please use the checklist as a guideline for the sort of information we need WITHIN the manuscript as well as in the checklist. This is particularly important for animal reporting, antibody dilutions (missing) and exact p-values and n that should be indicated instead of a range.

9) Every published paper now includes a 'Synopsis' to further enhance discoverability. Synopses are displayed on the journal webpage and are freely accessible to all readers. They include a short stand first (maximum of 300 characters, including space) as well as 2-5 one sentence bullet points that summarise the paper. Please write the bullet points to summarise the key NEW findings. They should be designed to be complementary to the abstract - i.e. not repeat the same text. We encourage inclusion of key acronyms and quantitative information (maximum of 30 words / bullet point). Please use the passive voice. Please attach these in a separate file or send them by email, we will incorporate them accordingly.

You are also welcome to suggest a striking image or visual abstract to illustrate your article. If you do please provide a jpeg file 550 px-wide x 400-px high.

10) A Conflict of Interest statement should be provided in the main text

11) Please note that we now mandate that all corresponding authors list an ORCID digital identifier. This takes <90 seconds to complete. We encourage all authors to supply an ORCID identifier, which will be linked to their name for unambiguous name identification.

Currently, our records indicate that the ORCID for your account is 0000-0001-7145-0533.

Please click the link below to modify this ORCID:
Link Not Available

12) The system will prompt you to fill in your funding and payment information. This will allow Wiley to send you a quote for the article processing charge (APC) in case of acceptance. This quote takes into account any reduction or fee waivers that you may be eligible for. Authors do not need to pay any fees before their manuscript is accepted and transferred to our publisher.

Graphs 800-1,200 DPI
Photos 400-800 DPI
Colour (only CMYK) 300-400 DPI"

*Additional important information regarding figures and illustrations can be found at <https://bit.ly/EMBOPressFigurePreparationGuideline>

***** Reviewer's comments *****

Referee #1 (Comments on Novelty/Model System for Author):

The questions asked by the study require the use of an intact animal as a model as they involve gene delivery to the inner ear in the aim of hearing restoration. The pharmacology behind the gene delivery and the adequacy of the optogenetic protein can only be assessed correctly such a model. This is an early pre-clinical study and the use of mouse as a model organism is adequate to the level of advancement towards clinical application.

Referee #1 (Remarks for Author):

The manuscript by Bali et al, examines application of vfChrimsonR after AAV mediated delivery to the inner ear of mice. The authors assess the impact of using a faster version of Chrimson vfChrimson over kinetically improved version fChrimson (and retrospectively to the blue shifted but faster Chronos). They explore the complex equation around the expression levels, membrane targeting and kinetics of optogenetic proteins and the observed biological activity. Although their downstream electrophysiological assessments are strong and appropriate for the questions asked, the quantitative parts regarding gene delivery and expression has some weaknesses. Moreover, there seems to be some missing links between the results obtained and the interpretation the results in terms of the relevance of their findings to clinical gene therapy. It would be useful to clarify these points prior to publication in EMBO Mol Med.

Major:

The authors state that the trafficking of vf-Chrimson to the plasma membrane is less efficient, based on the photocurrent density estimated in NG cells for vf-Chrimson which dropped to one fourth of that of f-Chrimson instead of the theoretically expected one half. Do they have any quantitative data to support this? The authors should provide quantitative data (such as qRT-PCR) comparing expression of the two proteins in side by side carefully monitored conditions. In the comparisons made between ES/TS version of vfChrimson in vivo, how did the authors ensure equal particle numbers of AAV were injected in each animal and how many of the particles reached the SGNs? Were the vectors tittered side by side? Were the efflux that occur during injections monitored? At the very least these important factors should be mentioned in the text in absence of a quantitative method such as RNAscope comparing the transcription levels between the different constructs in vivo.

The trafficking / entry signals used here seem to be derived from rodent sequences based on the cited literature. What is the homology between the rodent and equivalent human sequences? Do the authors think a human cell might make the best out of these signals in view of the downstream

clinical application? Should this type of design be changed when moving from preclinical to clinical studies? This point would be worthwhile to tackle in the discussion.

The manuscript would also benefit from the following minor revisions:

Avoid usage of subjective wording such as: pp4 last sentence- "a fair range of light intensities"
pp25 "Neural population responses of vf-Chrimson-expressing SGNs remained sizable for stimulation rates greater than 500 Hz." It would help the reader what is considered a fair range and what is a sizable response for instance in comparison to normal physiology.

On page 5- The injection method used in p6 animals should be stated as local injection - as a reader might expect a systemic injection based on the viral vectors chosen.

On Figure 2- A through C: it would help the reader to have the time to pulse onset values for both vfChrimson and vfChrimsonES/TS

Referee #2 (Comments on Novelty/Model System for Author):

This manuscript provides important data and insights for optogenetic hearing restoration. Overall, the study is well performed and I have only minor concerns/comments. Maybe the clarity for nonspecialists could be improved.

Referee #2 (Remarks for Author):

The work by Bali et al. studies the temporal fidelity as well as the intensity coding of optogenetic-evoked responses in SGN neurons by two optogenetic Chrimson variants (f-Chrimson or vf-Chrimson). Both variants show slower channel kinetics than Chronos. Still, it is very sound that f-Chrimson provides near physiological temporal fidelity of coding. Other biophysical parameters such as lower light intensity for stimulation and thereby decreased phototoxicity highlight that Chrimson variants are ideal therapeutic candidates for optogenetic hearing restoration. In general, the data is well supported by different experimental and analytical approaches and adds valuable knowledge towards clinical translation.

Minor comments:

Page 3: What does "retinal implantation" mean?

Page 5: What are "NG cells"?

Page 5: Are the same AAV capsids used for vf-Chrimson and vf-Chrimson-ES/TS?

Page 6: Figure Bii: The color codes for vf-Chrimson and vf-Chrimson-ES/TS are very similar. It would be more reader friendly and clearer to choose a more distinguishable colors.

Page 8: "The injected mice behaved normally as concluded from routine animal observation." This needs to be clarified: which behavioral observations are meant?

Page 11: Please summarize the differences between vf-Chrimson and vf-Chrimson-ES/TS not just between vf-Chrimson and f-Chrimson?

Page 14: The Color code makes the figures incomprehensible. Also, the plots are overcrowded and the information extraction is difficult.

Page 18: Please add x and y scales to all panels.

Page 22: Figure 6 A,B: x-axis not defined.

Page 31: Material and Methods: "Cloning" is lab jargon, please use more accurate heading. Overall, the clarity and English of the Material and Methods section is of less quality than the main text. A revision to improve the language and thereby the clarity would be helpful.

Referee #3 (Comments on Novelty/Model System for Author):

Work on optical cochlear implants is overall very new, with most work occurring in the past 5 years or so. This is a highly clinically relevant subject, and there are hopes for implantation of oCIs within a few years in humans. This is one of a few studies detailing potential improvements to the optogenetic proteins used in oCIs, but finding optogenetic variants with the speed and power to effectively encode auditory stimuli is critical to the clinical use of oCIs.

Referee #3 (Remarks for Author):

In the manuscript by Bali et al (EMM-2020-13391) the authors follow up on recent work from their group to introduce optogenetic proteins into cochlear spiral ganglion neurons for the purpose of creating optical cochlear implants. Despite the success of cochlear implants in restoring open speech comprehension in the majority of users, the technology is limited by the wide spread of current around each electrode contact which restricts the spectral resolution of sound coding. Optical stimulation via an optical CI (oCI) offers an alternative to electrical stimulation as long as neurons can be rendered sufficiently light sensitive. High temporal fidelity is imperative in order to preserve sound coding; this requires channelrhodopsins (ChR) with fast kinetics to allow spiral ganglion neurons (SGN) to fire at near physiological rates. f-Chrimson and its variants are candidates for optogenetic sensory restoration. This work investigates two ChR variants, f-Chrimson and vf-Chrimson for their suitability for optogenetic SGN stimulation, addressing temporal fidelity and intensity coding. The authors also investigate a vf-Chrimson modified with export signal and trafficking signals sequences in an attempt to enhance membrane trafficking for improved function.

The authors used histological techniques to demonstrate expression of the channelrhodopsin variants in SGN and localization to the cellular membrane, extracellular recordings from individual SGN axons to carefully characterize neuronal responses to light pulses (and sound stimuli as a control), and auditory brainstem responses (ABRs) evoked by both acoustic and optical stimuli to compare neuron function in vivo between control and ChR transduced animals. Adding the ES/TS signals to vf-Chrimson improved membrane localization but was not found to offer any benefit over vf-Chrimson. Analysis of temporal fidelity, radiant flux required for activation, and dynamic range of SGN coding at the single cell level demonstrated that while f-Chrimson has an improved dynamic range relative to electrical stimulation, (but worse dynamic range compared to acoustic activation), vf-Chrimson has improved temporal coding relative to f-Chrimson. Therefore, although there are still improvements to be made to achieve both the true coding speed and dynamic range of acoustic stimulation of the auditory nerve, these ChR variants demonstrate improved function in a number of measures relative to older ChR2 variants and bring us closer to clinical use of optical cochlear implants.

Overall the work is thorough, well analyzed, and well presented. Improved channelrhodopsin variants for enhanced function is necessary to clinical application of optical cochlear implants, and this manuscript gives a balanced assessment of current state-of-the-art variants (both f-Chrimson and vf-Chrimson), presenting improvements in function over previous variants as well as clearly discussing limitations and modifications that did not provide a functional benefit (i.e. addition of the trafficking signal). Some minor changes to be made to figures and text are detailed below.

Minor

Some figures (eg figure 4) need better contrast to distinguish between both individual and population data, as well as between the different genotypes.

Page 2 - line 10 change 'lend to a' to 'lend a'

Page 3 - line 15 '700.00' typo: to '700,000'

Page 4 - line 19 -syntax- remove the word 'translating' or add 'to clinic' or something similar at the end of the sentence

Page 6 - Figure 1 - panels D/E - make consistent in terms of labels on both graphs, and n

Page 7 - Figure 1 caption, lines 23-25 - The text says 'number of transduced SGNs seems higher in the apical and middle turns compared to the basal one for both opsin constructs', but this is hard to see, especially in the middle turn. 'Seems' is very imprecise, please detail quantification or remove the statement.

Figure 1C - There are fewer transduced cells in the high frequency cochlear base. Please address why, if known, and impact on oCI function.

Page 8 - line 5 - insert word 'cochlea' in front of '(Figure 1C)'

Page 10 - lines 10-11 - the inset box-and-whisker plot in panel H looks like it is missing portions of the plot, the data on the left has no whiskers and the data on the right has no box, is this an error?

Page 10 - line 22 - clarification needed- are you saying the radiant flux is changed over an order of magnitude? Also, this should refer to panel 2A, not 2B.

Page 11 - line 14 - "mediated"

Page 12 -figure 3B - why does the vf-Chrimson-ES/TS amplitude decrease more from the first to the 20th iteration more so than the vf-Chrimson?

Page 15 - lines 2-3 - the temporal fidelity is measures by two means, temporal jitter and vector strength. Temporal jitter is considerably less for f-Chrimson compared to vf-Chrimson at stimulation rates >200 Hz and so it could be concluded that it has a superior performance to vf-Chrimson. Perhaps restrict conclusion to lower spike rates.

Page 17 - line 12 - what is (pe)? I could not see it defined anywhere

Page 22 - line 2 - insert "of" after the word "trains"

Page 27 - line 4 - missing words in sentence "at least in part the greater"

Page 27 - line 10 - remove "also to"

Page 27 - line 18 - please add a unit for 2.5 (I assume it is X or times). Also, where did the figure 2.5 lower come from? The jitter in figures 6C and D is 3 vs 1.8 at the largest difference (at the lowest stim) at the higher stimulations this difference drops to .4

Page 33 - line 11 "as a pain reliever"

Page 33 - line 12 "tracked daily"

Page 33 - line 20 - "we planned to use" - was it used? If yes, rephrase

Page 34 - lines 16-18 - clarify how membrane/intracellular expression ratio areas were defined, it is not clear what the values are referring to, especially the "-11 um" value

Page 36 - line 4 rearrange - "20 Hz being"

Page 36 - line 6 wording "equal to or greater than"

Page 36 - line 9 -move "being" to before "cycle"

Referee #1 (Comments on Novelty/Model System for Author):

The questions asked by the study require the use of an intact animal as a model as they involve gene delivery to the inner ear in the aim of hearing restoration. The pharmacology behind the gene delivery and the adequacy of the optogenetic protein can only be assessed correctly such a model. This is an early pre-clinical study and the use of mouse as a model organism is adequate to the level of advancement towards clinical application.

Referee #1 (Remarks for Author):

We thank the reviewer for the appreciation of our work and the constructive criticism that helped us to improve the MS. We have addressed all concerns and have performed further experiments to further strengthen our study. Most importantly we have now studied vf-Chrimson-mediated intensity encoding by individual putative spiral ganglion neurons. In addition to the spike-rate based intensity encoding, we have also analyzed how temporal fidelity of encoding changes as a function of stimulus intensity for optogenetic and acoustic stimulation. Finally, we have also compared membrane targeting of vf-Chrimson and f-Chrimson in NG cells in order to address this reviewer's request. The revised work includes roughly 1/3 more data than the original submittal and has been carefully revised according to the reviewers' advice.

The manuscript by Bali et al, examines application of vfChrimsonR after AAV mediated delivery to the inner ear of mice. The authors assess the impact of using a faster version of Chrimson vfChrimson over kinetically improved version fChrimson (and retrospectively to the blue shifted but faster Chronos). They explore the complex equation around the expression levels, membrane targeting and kinetics of optogenetic proteins and the observed biological activity. Although their downstream electrophysiological assessments are strong and appropriate for the questions asked, the quantitative parts regarding gene delivery and expression has some weaknesses. Moreover, there seems to be some missing links between the results obtained and the interpretation the results in terms of the relevance of their findings to clinical gene therapy. It would be useful to clarify these points prior to publication in EMBO Mol Med.

Major:

The authors state that the trafficking of vf-Chrimson to the plasma membrane is less efficient, based on the photocurrent density estimated in NG cells for vf-Chrimson which dropped to one fourth of that of f-Chrimson instead of the theoretically expected one half. Do they have any quantitative data to support this? The authors should provide quantitative data (such as qRT-PCR) comparing expression of the two proteins in side by side carefully monitored conditions.

In response to the reviewer's comment we have performed further experiments and analyses. Specifically, we have analyzed NG cell expression of vf-Chrimson and f-Chrimson using GFP-immunofluorescence and line profiles in confocal sections. The results indicate three categories of cells: i) cells with a clear plasma membrane peak of immunofluorescence, ii) cells with comparable immunofluorescence of plasma membrane and intracellular space and iii) intracellular immunofluorescence outweighing that of the plasma membrane. Expression of f-Chrimson led to a larger fraction of NG108-15 cells with greater plasma membrane expression (9 out of 30 cells for f-Chrimson vs. 3 out of 30 cells for vf-Chrimson). Poorer plasma membrane expression likely explains why the photocurrent density mediated by vf-Chrimson was lower than for f-Chrimson. We have included this data as Expanded View Figure 1 and appended the corresponding results and discussion sections. There we also enhanced the discussion of

how neural light sensitivity relates to the membrane expression and the open time of the channel.

In the comparisons made between ES/TS version of vfChrimson in vivo, how did the authors ensure equal particle numbers of AAV were injected in each animal and how many of the particles reached the SGNs? Were the vectors tittered side by side? Were the efflux that occur during injections monitored? At the very least these important factors should be mentioned in the text in absence of a quantitative method such as RNAscope comparing the transcription levels between the different constructs in vivo.

We have performed all injections as described and identically for the specific viral vectors at the given titers. Titters were quantified by standardized qPCR and vectors stored froze at -80°C until use. Efflux was rarely observed during these injections. In most cases of injection of vf-Chrimson-ES/TS and vf-Chrimson AAVs, entire litters were injected with one viral vector for ease of work. However, the injections were done in the same time frame (within few weeks). Most of the vf-Chrimson work presented here was done after that on f-Chrimson, but the injections were performed by the same personnel following the same procedure. In response to the reviewer's comment we have strengthened the corresponding sections of materials and methods, as well as of results. Moreover, we have added a sentence to discussion to mention that differences in titer and viral vector as well as trial to trial variability of the AAV injections might contribute to the variance in the functional responses, e.g. of oABR.

"More generally, we note that small differences in titer and viral vector as well as trial to trial variability of the AAV-injection might contribute to the variance in the functional responses such as oABR."

The trafficking / entry signals used here seem to be derived from rodent sequences based on the cited literature. What is the homology between the rodent and equivalent human sequences? Do the authors think a human cell might make the best out of these signals in view of the downstream clinical application? Should this type of design be changed when moving from preclinical to clinical studies? This point would be worthwhile to tackle in the discussion.

We thank the reviewer for this valuable comment. At this point while the DNAs of vf-Chrimson and EYFP were optimized for human codon usage, the ER export and trafficking sequence was taken from mouse Kir2.1 following Hofherr et al., 2005 and Gradinaru et al., 2010. We have further stressed the need for future preclinical trials to "humanized" and least complex constructs for enhancing biosafety of optogenetic SGN manipulation.

"These preclinical efforts should also aim for using trafficking sequences of human membrane proteins (such as Kir2.1) and evaluate the effect of omitting the fluorescent protein from the expression construct to enhance biosafety." done

The manuscript would also benefit from the following minor revisions:

Avoid usage of subjective wording such as: pp4 last sentence- "a fair range of light intensities" pp25 "Neural population responses of vf-Chrimson-expressing SGNs remained sizable for stimulation rates greater than 500 Hz." It would help the reader what is considered a fair range and what is a sizable response for instance in comparison to normal physiology.

done

On page 5- The injection method used in p6 animals should be stated as local injection - as a reader might expect a systemic injection based on the viral vectors chosen.

done

On Figure 2- A through C: it would help the reader to have the time to pulse onset values for both vfChrimson and vfChrimsonES/TS

We are not sure to fully comprehend the comment. Is it possible that the reviewer is concerned about the negative deflection of the signal prior to pulse onset: this is due to low-pass filtering of the oABR. In response to the comment we have relabeled the axis as "time" and mentioned the filter effect in the results section.

Referee #2 (Comments on Novelty/Model System for Author):

This manuscript provides important data and insights for optogenetic hearing restoration. Overall, the study is well performed and I have only minor concerns/comments. Maybe the clarity for nonspecialists could be improved.

Referee #2 (Remarks for Author):

The work by Bali et al. studies the temporal fidelity as well as the intensity coding of optogenetic-evoked responses in SGN neurons by two optogenetic Chrimson variants (f-Chrimson or vf-Chrimson). Both variants show slower channel kinetics than Chronos. Still, it is very sound that f-Chrimson provides near physiological temporal fidelity of coding. Other biophysical parameters such as lower light intensity for stimulation and thereby decreased phototoxicity highlight that Chrimson variants are ideal therapeutic candidates for optogenetic hearing restoration. In general, the data is well supported by different experimental and analytical approaches and adds valuable knowledge towards clinical translation.

We thank the reviewer for the appreciation of our work and the constructive criticism that helped us to improve the MS. We have addressed all concerns and have performed further experiments to further strengthen our study. Most importantly we have now studied vf-Chrimson-mediated intensity encoding by individual putative spiral ganglion neurons. In addition to the spike-rate based intensity encoding, we have also analyzed how temporal fidelity of encoding changes as a function of stimulus intensity for optogenetic and acoustic stimulation. Finally, we have also compared membrane targeting of vf-Chrimson and f-Chrimson in NG cells in order to address a reviewer's request. The revised work includes roughly 1/3 more data than the original submittal and has been carefully been revised according to the reviewers' advice.

Minor comments:

Page 3: What does "retinal implantation" mean?

Fixed: "As a matter of fact, production of retinal implants has recently stopped in Europe and the US."

Page 5: What are "NG cells"?

NG108-15 cells (ATCC, HB-12377TM, Manassas, USA) represent a hybrid neuroblastoma and neuroglioma cell line, Fixed!

Page 5: Are the same AAV capsids used for vf-Chrimson and vf-Chrimson-ES/TS?

No, two potent AAV9 derived capsids generated by the Gradinaru lab were used as stated "AAV-PHP.B (vf-Chrimson) or AAV-PHP.eB (vf-Chrimson-ES/TS)". We note, that we did not find significant differences in the transduction rate (Fig. 1D), hence, we believe that the subtle differences can be attributed to the ER-exit and trafficking signals in vf-Chrimson-ES/TS. In response to the reviewer's comment we have added a note of caution in the discussion.

"...potent viral vectors (AAV-PHP.B and AAV-PHP.eB with transduction rates being statistically indistinguishable from each other).."

Page 6: Figure Bii: The color codes for vf-Chrimson and vf-Chrimson-ES/TS are very similar. It would be more reader friendly and clearer to choose a more distinguishable colors.

Done

Page 8: "The injected mice behaved normally as concluded from routine animal observation."
This needs to be clarified: which behavioral observations are meant?
Done, we have specified the notion. This refers to lack of phenotypes obvious to scientists and animal caretakers without use of technical means and includes epilepsy, circling, abnormal motor activity, reduced body size etc.

Page 11: Please summarize the differences between vf-Chrimson and vf-Chrimson-ES/TS not just between vf-Chrimson and f-Chrimson?

We are a bit uncertain what is meant: all text on page 11 compares vf-Chrimson and vf-Chrimson-ES/TS. In response to the reviewer's comment we have tried to enhance the summary statement for clarity:

"In summary, the oABR comparison indicates greater temporal fidelity but higher required radiant flux for both vf-Chrimson-ES/TS and vf-Chrimson-mediated SGN stimulation than previously found for f-Chrimson (Mager et al, 2018)."

Page 14: The Color code makes the figures incomprehensible. Also, the plots are overcrowded and the information extraction is difficult.

Done, in response to the comment, we have separated the vf-Chrimson-ES/TS and vf-Chrimson data and only replot the mean \pm SEM for comparison.

Page 18: Please add x and y scales to all panels.

Done

Page 22: Figure 6 A,B: x-axis not defined.

Done

Page 31: Material and Methods: "Cloning" is lab jargon, please use more accurate heading.

Done

Overall, the clarity and English of the Material and Methods section is of less quality than the main text. A revision to improve the language and thereby the clarity would be helpful.

We apologize and have carefully revised the methods section.

Referee #3 (Comments on Novelty/Model System for Author):

Work on optical cochlear implants is overall very new, with most work occurring in the past 5 years or so. This is a highly clinically relevant subject, and there are hopes for implantation of oCIs within a few years in humans. This is one of a few studies detailing potential improvements to the optogenetic proteins used in oCIs, but finding optogenetic variants with the speed and power to effectively encode auditory stimuli is critical to the clinical use of oCIs.

Referee #3 (Remarks for Author):

In the manuscript by Bali et al (EMM-2020-13391) the authors follow up on recent work from their group to introduce optogenetic proteins into cochlear spiral ganglion neurons for the purpose of creating optical cochlear implants. Despite the success of cochlear implants in restoring open speech comprehension in the majority of users, the technology is limited by the wide spread of current around each electrode contact which restricts the spectral resolution of sound coding. Optical stimulation via an optical CI (oCI) offers an alternative to electrical stimulation as long as neurons can be rendered sufficiently light sensitive. High temporal fidelity is imperative in order to preserve sound coding; this requires channelrhodopsins (ChR) with fast kinetics to allow spiral ganglion neurons (SGN) to fire at near physiological rates. f-Chrimson and its variants are candidates for optogenetic sensory restoration. This work investigates two ChR variants, f-Chrimson and vf-Chrimson for their suitability for optogenetic SGN stimulation, addressing temporal fidelity and intensity coding. The authors also investigate a vf-Chrimson modified with export signal and trafficking signals sequences in an attempt to enhance membrane trafficking for improved function.

The authors used histological techniques to demonstrate expression of the channelrhodopsin variants in SGN and localization to the cellular membrane, extracellular recordings from individual SGN axons to carefully characterize neuronal responses to light pulses (and sound stimuli as a control), and auditory brainstem responses (ABRs) evoked by both acoustic and optical stimuli to compare neuron function in vivo between control and ChR transduced animals. Adding the ES/TS signals to vf-Chrimson improved membrane localization but was not found to offer any benefit over vf-Chrimson. Analysis of temporal fidelity, radiant flux required for activation, and dynamic range of SGN coding at the single cell level demonstrated that while f-Chrimson has an improved dynamic range relative to electrical stimulation, (but worse dynamic range compared to acoustic activation), vf-Chrimson has improved temporal coding relative to f-Chrimson. Therefore, although there are still improvements to be made to achieve both the true coding speed and dynamic range of acoustic stimulation of the auditory nerve, these ChR variants demonstrate improved function in a number of measures relative to older ChR2 variants and bring us closer to clinical use of optical cochlear implants.

Overall the work is thorough, well analyzed, and well presented. Improved channelrhodopsin variants for enhanced function is necessary to clinical application of optical cochlear implants, and this manuscript gives a balanced assessment of current state-of-the-art variants (both f-Chrimson and vf-Chrimson), presenting improvements in function over previous variants as well as clearly discussing limitations and modifications that did not provide a functional benefit (i.e. addition of the trafficking signal). Some minor changes to be made to figures and text are detailed below.

We thank the reviewer for the appreciation of our work and the constructive criticism that helped us to improve the MS. We have addressed all concerns and have performed further experiments to further strengthen our study. Most importantly we have now studied vf-Chrimson-mediated intensity encoding by individual putative spiral ganglion neurons. In addition to the spike-rate based intensity encoding, we have also analyzed how temporal fidelity of encoding changes as a function of stimulus intensity for optogenetic and acoustic stimulation. Finally, we have also compared membrane targeting of vf-Chrimson and f-Chrimson in NG cells in order to address a reviewer's request. The revised work includes roughly 1/3 more data than the original submittal and has been carefully been revised according to the reviewers' advice.

Minor

Some figures (eg figure 4) need better contrast to distinguish between both individual and population data, as well as between the different genotypes.

Done

Page 2 - line 10 change 'lend to a' to 'lend a'

Done

Page 3 - line 15 '700.00' typo: to '700,000'

Done

Page 4 - line 19 -syntax- remove the word 'translating' or add 'to clinic' or something similar at the end of the sentence

Done

Page 6 - Figure 1 - panels D/E - make consistent in terms of labels on both graphs, and n

Done

Page 7 - Figure 1 caption, lines 23-25 - The text says 'number of transduced SGNs seems higher in the apical and middle turns compared to the basal one for both opsin constructs', but this is hard to see, especially in the middle turn. 'Seems' is very imprecise, please detail quantification or remove the statement.

Done

Figure 1C - There are fewer transduced cells in the high frequency cochlear base. Please address why, if known, and impact on oCI function.

Done

"Clearly, incomplete and inhomogeneous SGN transduction along the tonotopic axis as well as viral spread indicate the need for further optimization of AAV-mediated gene transfer for improved efficiency and specificity."

Page 8 - line 5 - insert word 'cochlea' in front of '(Figure 1C)'

Done

Page 10 - lines 10-11 - the inset box-and-whisker plot in panel H looks like it is missing portions of the plot, the data on the left has no whiskers and the data on the right has no box, is this an error?

We agree that this whisker plot looks unusual but it reflects the fact that for vf-Chrimson-ES/TS 8 of 9 mice had a maximum oABR amplitude in response to 0.4 ms light pulse, and another one at 0.6.

Page 10 - line 22 - clarification needed- are you saying the radiant flux is changed over an order of magnitude? Also, this should refer to panel 2A, not 2B.

Done

Page 11 - line 14 - "mediated"

Done

Page 12 -figure 3B - why does the vf-Chrimson-ES/TS amplitude decrease more from the first to the 20th iteration more so than the vf-Chrimson?

We note that these are the responses of exemplary SGNs and while we aim to show representative responses, this aspect is not, as one can appreciate from Fig. 4C. We have noted this in the corresponding text: change from "representative" to "exemplary" for Fig. 3, Spike probability declined in both cases "in a similar manner."

Page 15 - lines 2-3 - the temporal fidelity is measures by two means, temporal jitter and vector strength. Temporal jitter is considerably less for f-Chrimson compared to vf-Chrimson at stimulation rates >200 Hz and so it could be concluded that it has a superior performance to vf-Chrimson. Perhaps restrict conclusion to lower spike rates.

Done

Page 17 - line 12 - what is (pe)? I could not see it defined anywhere

SPL (pe) "peak equivalent sound pressure level" for reporting sound pressure level for clicks

Page 22 - line 2 - insert "of" after the word "trains"

Done

Page 27 - line 4 - missing words in sentence "at least in part the greater"

Done

Page 27 - line 10 - remove "also to"

Done

Page 27 - line 18 - please add a unit for 2.5 (I assume it is X or times). Also, where did the figure 2.5 lower come from? The jitter in figures 6C and D is 3 vs 1.8 at the largest difference (at the lowest stim) at the higher stimulations this difference drops to .4

The value of ~2.5 times smaller comes from dividing the jitter for clicks and light stimuli at maximal stimulation intensity: $1.06/0.39 = 2.7$

In response to the reviewer's comment we have kept the statement of significance (reported in results) and removed the fold difference as it is not critical.

Page 33 - line 11 "as a pain reliever"

Done

Page 33 - line 12 "tracked daily"

Done

Page 33 - line 20 - "we planned to use" - was it used? If yes, rephrase

Done

Page 34 - lines 16-18 - clarify how membrane/intracellular expression ratio areas were

defined, it is not clear what the values are referring to, especially the "-11 um" value
Done

Page 36 - line 4 rearrange - "20 Hz being"
Done

Page 36 - line 6 wording "equal to or greater than"
Done

Page 36 - line 9 -move "being" to before "cycle"
Done

26th Feb 2021

Thank you for the submission of your revised manuscript to EMBO Molecular Medicine. We have now received the enclosed report from the two referees who were asked to re-assess it. As you will see the referees are now supportive and I am pleased to inform you that we will be able to accept your manuscript pending the following amendments:

Please address the minor concerns raised by Referee #3.

On a more editorial level, please do the following:

1. in the main manuscript file:

-remove the red color font.

-the callouts for Figure 2I, Figure 3A, 3B, Figure 5D, 5E, Figure 7I-L are missing. Please fix this.

- the main manuscript file should not contain figures. Figures should be uploaded separately. Please provide individual production quality figure files as .eps, .tif, .jpg (one file per figure). Figure legends should be included in the manuscript .doc file.

2. Please rename 'competing interest' to "Conflicts of interest".

According to our editorial policy with regard to the "conflict of interest" (see below), the current statement suggests that you have no specific financial interest to declare - please confirm that.

'the journal requires authors of original research papers to declare any competing commercial interests in relation to the submitted work. It is difficult to specify a threshold at which a financial interest becomes significant, but as a practical guideline, we would suggest this to be any undeclared interest that could embarrass you were it to become publicly known.'

<https://www.embopress.org/page/journal/17574684/authorguide#conflictsofinterest>

Please also add 'The other authors declare no conflict of interests.'

3. Data Availability: since this study does not generate large-scale datasets, please only include the following sentence in this section- "This study includes no data deposited in external repositories".

4. Author checklist: please enter all three corresponding authors' names.

5. The Paper Explained: EMBO Molecular Medicine articles are accompanied by a summary of the articles to emphasize the major findings in the paper and their medical implications for the non-specialist reader. Please provide a draft summary of your article highlighting

a. the medical issue you are addressing,

b. the results obtained and

c. their clinical impact.

6. For More Information: There is space at the end of each article to list relevant web links for further

consultation by our readers. Could you identify some relevant ones and provide such information as well? Some examples are patient associations, relevant databases, OMIM/proteins/genes links, author's websites, etc..

7. Every published paper now includes a 'Synopsis' to further enhance discoverability. Synopses are displayed on the journal webpage and are freely accessible to all readers. They include a short stand first (maximum of 300 characters, including space) as well as 2-5 one sentence bullet points that summarize the paper. Please write the bullet points to summarize the key NEW findings. They should be designed to be complementary to the abstract - i.e. not repeat the same text. We encourage inclusion of key acronyms and quantitative information (maximum of 30 words / bullet point). Please use the passive voice. Please attach these in a separate file or send them by email, we will incorporate them accordingly.

8. Please provide a "synopsis image" or visual abstract (550px width and 400px height, jpeg format) to highlight the paper on our homepage.

9. We would also encourage you to include the source data for figure panels that show essential data. Numerical data should be provided as individual .xls or .csv files (including a tab describing the data). For blots or microscopy, uncropped images should be submitted (using a zip archive if multiple images need to be supplied for one panel). Additional information on source data and instruction on how to label the files are available at <https://www.embopress.org/page/journal/17574684/authorguide#sourcedata>

10. Our data editor has made a couple of suggestions about your manuscript (see attached file). Please address these issues and keep the track mode on.

11. As part of the EMBO Publications transparent editorial process initiative (see our Editorial at <http://embomolmed.embopress.org/content/2/9/329>), EMBO Molecular Medicine will publish online a Review Process File (RPF) to accompany accepted manuscripts.

a. In the event of acceptance, this file will be published in conjunction with your paper and will include the anonymous referee reports, your point-by-point response and all pertinent correspondence relating to the manuscript. Let us know if you do NOT agree with this.

I look forward to seeing a revised version of your manuscript as soon as possible.

Sincerely,
Jingyi

Jingyi Hou
Editor
EMBO Molecular Medicine

*** Instructions to submit your revised manuscript ***

To submit your manuscript, please follow this link:

Link Not Available

- 1) a .docx formatted version of the manuscript text (including Figure legends and tables)
- 2) Separate figure files*
- 3) supplemental information as Expanded View and/or Appendix. Please carefully check the authors guidelines for formatting Expanded view and Appendix figures and tables at <https://www.embopress.org/page/journal/17574684/authorguide#expandedview>
- 4) a letter INCLUDING the reviewer's reports and your detailed responses to their comments (as Word file).
- 5) The paper explained: EMBO Molecular Medicine articles are accompanied by a summary of the articles to emphasize the major findings in the paper and their medical implications for the non-specialist reader. Please provide a draft summary of your article highlighting
 - the medical issue you are addressing,
 - the results obtained and
 - their clinical impact.This may be edited to ensure that readers understand the significance and context of the research. Please refer to any of our published articles for an example.
- 6) For more information: There is space at the end of each article to list relevant web links for further consultation by our readers. Could you identify some relevant ones and provide such information as well? Some examples are patient associations, relevant databases, OMIM/proteins/genes links, author's websites, etc...
- 7) Author contributions: the contribution of every author must be detailed in a separate section.
- 8) EMBO Molecular Medicine now requires a complete author checklist (<https://www.embopress.org/page/journal/17574684/authorguide>) to be submitted with all revised manuscripts. Please use the checklist as guideline for the sort of information we need WITHIN the manuscript. The checklist should only be filled with page numbers where the information can be found. This is particularly important for animal reporting, antibody dilutions (missing) and exact

values and n that should be indicated instead of a range.

9) Every published paper now includes a 'Synopsis' to further enhance discoverability. Synopses are displayed on the journal webpage and are freely accessible to all readers. They include a short stand first (maximum of 300 characters, including space) as well as 2-5 one sentence bullet points that summarise the paper. Please write the bullet points to summarise the key NEW findings. They should be designed to be complementary to the abstract - i.e. not repeat the same text. We encourage inclusion of key acronyms and quantitative information (maximum of 30 words / bullet point). Please use the passive voice. Please attach these in a separate file or send them by email, we will incorporate them accordingly.

You are also welcome to suggest a striking image or visual abstract to illustrate your article. If you do please provide a jpeg file 550 px-wide x 400-px high.

10) A Conflict of Interest statement should be provided in the main text

11) Please note that we now mandate that all corresponding authors list an ORCID digital identifier. This takes <90 seconds to complete. We encourage all authors to supply an ORCID identifier, which will be linked to their name for unambiguous name identification.

Currently, our records indicate that the ORCID for your account is 0000-0001-7145-0533.

Link Not Available

12) The system will prompt you to fill in your funding and payment information. This will allow Wiley to send you a quote for the article processing charge (APC) in case of acceptance. This quote takes into account any reduction or fee waivers that you may be eligible for. Authors do not need to pay any fees before their manuscript is accepted and transferred to our publisher.

Photos 400-800 DPI

*Additional important information regarding figures and illustrations can be found at <https://bit.ly/EMBOPressFigurePreparationGuideline>

The system will prompt you to fill in your funding and payment information. This will allow Wiley to send you a quote for the article processing charge (APC) in case of acceptance. This quote takes into account any reduction or fee waivers that you may be eligible for. Authors do not need to pay

any fees before their manuscript is accepted and transferred to our publisher.

***** Reviewer's comments *****

Referee #2 (Remarks for Author):

I would like to thank the authors for thoroughly addressing all my points within the revised manuscript, which is now very clear, comprehensive and sound.

Referee #3 (Comments on Novelty/Model System for Author):

This work details the testing of new variants of optogenetic proteins for use in optical cochlear implants, which are highly clinically relevant for thousands of individuals receiving hearing device implants. The use of optogenetic proteins to stimulate the auditory pathways has been shown by this group before, but this work details the potential of faster variants that will be better able to encode sound stimuli with high temporal fidelity.

Referee #3 (Remarks for Author):

In the revised manuscript by Bali et al (EMM-2020-13391-V2) the authors have addressed our major concerns with the work. They have added new data, revised analyses, and greatly improved figure clarity.

With the large amount of added text, there are a few additional typos and suggested edits:

Figure legend 1 and results text - inconsistent use of GFP and YFP

Page 11 - line 13 - "membrane(EV1E-F)" needs a space between membrane and parenthesis

Figure 2D f-Chrimson is referred to in the legend as 'brown', should be green

Page 23 - line 9 - crossed out 'mW' but not '38' so it will appear as 3843 mW in simple markup

Page 28 - line 5 - typo - 'replotted'

Figure 4 legend - define hazard function

Page 40 line 18 - typo - 'intenties' to 'intensities'

Page 42 - 5D and 5E not referenced, perhaps 5C listed instead

Page 43 - line 1 - parenthesis missing

Page 43 - line 4 - 'monoticity' to 'monotonicity'

Page 43 - line 13 - 'Next, we quantified the adaptation ratio' - text refers to Fig 6L but adaptation ratio is shown in 6J

Page 49 - line 18 - refers to Fig 7E-H when describing VS. VS is depicted in I-L

Page 49 - line 21 - refers to Fig 7E-H when describing insets in I-L

Page 53 - line 5 - 'Inserts in (A-D)' is referring to inserts in I-L

Line 72 - Line 11 - rasion to ratio

The authors performed the requested editorial changes.

23rd Mar 2021

We are pleased to inform you that your manuscript is accepted for publication and is now being sent to our publisher to be included in the next available issue of EMBO Molecular Medicine.

We would like to remind you that as part of the EMBO Publications transparent editorial process initiative, EMBO Molecular Medicine will publish a Review Process File online to accompany accepted manuscripts. If you do NOT want the file to be published or would like to exclude figures, please immediately inform the editorial office via e-mail.

Please read below for additional IMPORTANT information regarding your article, its publication and the production process.

Congratulations on your interesting work,
Jingyi

Jingyi Hou
Editor
EMBO Molecular Medicine

Follow us on Twitter @EmboMolMed
Sign up for eTOCs at embopress.org/alertsfeeds

YOU MUST COMPLETE ALL CELLS WITH A PINK BACKGROUND ↓
PLEASE NOTE THAT THIS CHECKLIST WILL BE PUBLISHED ALONGSIDE YOUR PAPER

Corresponding Author Name: Vladan Rankovic, Antoine Huet, Tobias Moser

Manuscript Number: EMM-2020-13391-V2